# Return to Work after Primary Total Knee Arthroplasty: The First Polish Pilot Retrospective Study

**DOI:** 10.3390/jcm13071902

**Published:** 2024-03-25

**Authors:** Mariusz Drużbicki, Łucja Kitrys, Jarosław Jabłoński, Damian Filip, Lidia Perenc, Agnieszka Guzik

**Affiliations:** Medical College, University of Rzeszów, 35-959 Rzeszów, Poland; mdruzb@ur.edu.pl (M.D.); lucja.kitrys6@gmail.com (Ł.K.); jajablonski@ur.edu.pl (J.J.); lperenc@ur.edu.pl (L.P.)

**Keywords:** knee, total knee arthroplasty, return to work

## Abstract

(1) **Background**: Total knee arthroplasty (TKA) performed on working-age patients significantly affects the participation of such patients in social life. A retrospective study was conducted to determine the return to work (RTW) rate after TKA. The goal of this study was to provide reference data for the Polish population and identify the factors impacting patients’ decisions to return to or resign from work, relative to their functional performance. (2) **Methods**: This retrospective study involved 48 patients. An interview related to RTW was carried out to identify the factors impacting a patient’s decision to return to or resign from work. Functional performance was assessed using the Knee Outcome Survey–Activities of Daily Living (KOS-ADL) scale. (3) **Results**: Before TKA, 15 individuals (31.25%) qualified for the study did not work and were receiving welfare benefits. After the surgery, 23 individuals (47.9% of those working prior to TKA) did not return to work. The number of those who did not work after TKA increased to 38 (79.17%), which was a significant change. The mean level of functional performance after TKA assessed using KOS-ADL was 75.89. (4) **Conclusions**: The findings show that the rate of RTW after TKA in Poland is significantly lower than that in other countries. The reasons for this situation, as shown in the study, may be related to the lack of an occupational rehabilitation system, resulting in a paucity of information about the possibility to return to work and about opportunities for retraining.

## 1. Introduction

Knee osteoarthritis is a concern for increasingly more people. This affliction leads to significant impairment of functional capacities, which adversely affects activity and participation, gradually leading to a decrease in the quality of life [1,2]. Total knee arthroplasty (TKA) is the most commonly applied intervention. This treatment is highly effective since it relieves pain and restores the function of the joint, consequently enabling improvements in the patients’ quality of life [3,4]. The success of the intervention is often measured by how well a person can resume their normal activities, and this may be linked to TKA kinematics and biomechanics. Poor kinematics and biomechanics after surgery can lead to issues such as pain, instability, and decreased range of motion, which can hinder a person’s capability to perform various activities effectively. In contrast, proper alignment and functioning of the implant can improve a person’s mobility and performance, increasing the likelihood of a successful recovery of various functions necessary for social participation [5,6,7,8,9,10,11]. Although numerous studies have shown that TKA is one of the most effective procedures in orthopedics [12,13,14,15,16,17,18], the outcomes can vary based on several factors, such as patient health status, including mental health, as well as expectations, comorbidities, age, preoperative pain ratings, preoperative levels of physical activity, and rehabilitation adherence [19,20,21,22,23,24,25,26]. 

The effectiveness of TKA can also be evaluated taking into account patients’ ability to return to work (RTW). In 2014, Tilbury in a systematic review of the related literature showed that the RTW rate at 3 to 6 months after TKA ranged between 71% and 83% [27]. Another systematic review, by Van Leemput, showed that the RTW rate after TKA ranged from 36% to 89%. Those who worked before and returned to work after TKA accounted for 40% to 98% of the populations studied [28]. Given these significant differences in the reported statistics, it seems necessary to identify the factors impacting patients’ decisions about RTW after TKA. The study by Styron showed that a faster RTW after TKA was linked to female gender, self-employment, better postoperative physical and mental health outcomes, a higher functional comorbidity index, and an accessible workplace, whereas a slower RTW after TKA was related to the preoperative level of pain, more physically demanding work, and employee compensation [29]. Furthermore, according to Bardgett, the RTW process after TKA was affected by delays in surgical intervention, limited and often inconsistent RTW-related advice from healthcare professionals, and a lack of rehabilitation to optimize the patient’s recovery and facilitate RTW [30]. According to Van Leemput, a slower or no RTW after TKA was more likely to be observed in patients performing jobs of a more physical nature or among those who were absent from work before the surgery [28]. Leichtenberg established that a more advanced age, lower educational level, and preoperative absence from work were associated with a partial or no return to work [31]. A multivariate analysis of data collected from 289 patients by Scott et al. showed that heavy or moderate physical work and age were the main factors contributing to a return to employment of any kind or a return to the same job. In the group of patients working before TKA, all those younger than 50 years of age and 60% of those aged 50–54 years returned to work in some capacity [32]. 

According to the data from the National Health Fund, in Poland, as many as 37,821 knee replacement procedures were performed in 2020, including 30,615 TKA procedures. On average, the female and male patients were aged 69 and 67 years, respectively. Approximately 50% of those who received a knee replacement were less than 69 years of age [33]. Unfortunately, there are no precise official data regarding the TKA rates in the working age population in Poland. By comparison, the data published by the United Kingdom National Joint Registry show that, in 2016, patients up to 65 years of age accounted for 40% of cases [34]. Notably, the number of patients with diagnosed knee osteoarthritis is consistently growing. Similarly, the number of TKA procedures performed annually is increasing, and individuals up to 65 years of age are expected to be the largest group among those receiving this treatment [35]. 

In view of the above, and considering the fact that the duration of people’s working lives is increasing, RTW after TKA is an issue of great importance. However, in Poland, there are no reliable data concerning this subject matter. For this reason, the present retrospective study was designed to determine the RTW rates after TKA in a Polish population, in order to provide reference data, and identify the factors impacting patients’ decisions to return to or resign from work, as well as their functional capacities following TKA. It was hypothesized that RTW rates in patients after TKA in Poland would be similar to those in developed countries.

## 2. Materials and Methods

### 2.1. Participants

This retrospective, single-center study was conducted on a group of patients treated at the Department of Orthopedics and Traumatology of the Regional Clinical Hospital No. 2 in Rzeszów, Poland, between 2013 and 2018. All surgical procedures were performed by the same team of five orthopedists using the same surgical technique: tibia first and mechanical alignment (TF&MA). 

A retrospective analysis of the medical records was performed, taking into account patients meeting eligibility criteria, i.e., individuals with unilateral TKA, women aged ≤54 years, and men aged ≤59 years on the day of the surgery. These age criteria for men and women were applied because of the legal retirement age in Poland, which is 60 for women and 65 for men. It was assumed that a period of five years before reaching retirement age and a period of at least one year following TKA would be sufficient for the patients to return to work after TKA. The exclusion criteria applied in the study included revision knee arthroplasty and simultaneous bilateral TKA, other musculoskeletal diseases (e.g., spinal stenosis), a history of other surgeries or internal diseases making it impossible to return to work, oncological diseases, musculoskeletal infections, and lack of consent to participate. A lack of functional independence on the day of the examination was an excluding factor. The participant’s Body Mass Index (BMI) was calculated based on his/her body weight and height on the day of the examination. The BMI value did not impact the participants’ eligibility to be enrolled for the study [36,37]

Because of their similar construction design, kinematics, instrumentation, and corresponding surgical techniques, prostheses from two companies were selected: Columbus™ (Aesculap B Braun, Tuttlingen, Germany) (*n* = 30) and Vanguard (Biomet, Warsaw, IN, USA) (*n* = 18). Both designs are primary, condylar prostheses with over 15 years of orthopedic application and include unconstrained posterior cruciate retaining (CR). In addition, intact PCL provides stability under flexion and allows for femoral rollback.

Polyethylene–metal bearings were applied in all cases, and the same approach was used, i.e., the medial parapatellar approach with classic instrumentation, extracortical fixation for tibia first, and intramedullary femoral instrumentation. The same surgical technique for insertion and theory of component fixation were also applied. All procedures were preceded by preoperative planning in the mediCAD-6.0 software (mediCAD-6.0 Hectec GmbH, Landshut, Germany). No patelloplasty was performed.

In the total group of 431 patients after TKA, the inclusion criteria were met by 48 patients, including 23 females and 25 males. On average, the participants were aged 59 (±4.09) years.

All patients gave their consent to participate in the study. The study was approved by the Bioethics Commission of the Center for Post-graduate Medical Training in Warsaw (No. 71/PB/2012).

### 2.2. Measures

All patients were selected based on a review of medical records. These patients were approached via phone by a doctor, a member of the hospital staff, and the research team. A phone interview based on a questionnaire was conducted with all participants and led by a member of the team. The 14-item questionnaire was designed by the team of researchers (orthopedists and experienced physiotherapists). The first part of the questionnaire was related to the demographics and body size measurements. The subsequent items asked about the date of surgery, postoperative rehabilitation, and postoperative complications. The final part focused on the participants’ occupational status before and after TKA, factors that influenced the decision to return to work, and the self-assessment of health.

The patients were asked to assess their health status on a 5-point Likert-type scale by responding to the following question: “Which term most accurately describes your current health compared to others in the same age group?” Possible responses were as follows: very good (=1), good (=2), moderate (=3), not good (=4), or poor (=5). The scoring system was inverted to make the results easier to read (higher scores indicated better self-rated health) [38].

Functional limitations in everyday life due to a knee dysfunction following TKA were assessed using the Knee Outcome Survey–Activities of Daily Living (KOS-ADL) scale [39]. This survey is a reliable and accurate instrument that responds to changes in the functional performance of patients with various knee conditions who receive physiotherapy or orthopedic interventions [40,41]. The study applied a Polish version of KOS-ADL [42], and responses were given on a scale of 0–5. The maximum final score was 70 points. To calculate the Activity in Daily Living (ADL) index, the final score was divided by 70 and multiplied by 100. KOS-ADL scores were presented for values normalized to a range of 0–100.

### 2.3. Statistical Methods

Prior to the recruitment procedure (a retrospective analysis of medical records), the analysis of power was computed using Statistica, version 13.1 (StatSoft, Cracow, Poland). The general indicator of effect size for all outcome measures and the power of the study were 0.5, the probability was 0.05, and the maximum error was 10%. The estimated target sample size was 45. Ultimately, 48 participants were enrolled in this study.

Statistical analyses were computed using SPSS (IBM, Chicago, IL, USA), version 21.0. Categorical variables were reported as percentage values, and continuous variables were presented as the mean, standard deviation, and median values. Analysis of the normal distribution was carried out using a Shapiro–Wilk test. Comparative analysis of the KOS-ADL scores after surgery in the occupational activity groups was performed using the non-parametric Mann–Whitney test. The Spearman correlation coefficient was used to assess the relationship between functional performance measured with KOS-ADL and self-rated health status. Statistical significance was confirmed at *p* < 0.05.

## 3. Results

### 3.1. The Flow of Participants

Based on medical records, 431 patients were selected for this study. The age criterion was exceeded by 85.6% of the patients (369 individuals). Following the first discussion by phone, fourteen individuals were excluded (three individuals had died, three patients refused to participate, two individuals changed their phone number, and six patients reported that they had another TKA). The detailed flow of participants in the study is shown in Figure 1.

### 3.2. Characteristics of the Participants

Out of all the participants, 28 patients (58.3%) lived in rural areas and 20 patients (41.7%) lived in urban areas. The participants’ BMI ranged from 20.8 to 41.1. The mean BMI was 31.446 (median value of 31.947). Postoperative complications were reported by 83% of the participants, and all patients reported only intermittent pain in the lower limb that was operated upon. In descending order of frequency, the participants’ self-reported health was average (41.7%), good (27.1%), poor (16.7%), very good (12.5%), and excellent (2.1%). The occupational activities of the participants before and after TKA are shown in Table 1.

Complications after TKA, including restricted mobility of the knee and pain in the lower limb that was operated on, were reported on the day of the examination by 8.3% of the participants. In the assessment of current health, average health status was reported by 41.7%, good health by 27.1%, and poor health by 16.7% of participants; only 14.6% of the study participants reported very good health.

### 3.3. Factors Impacting a Decision to Resign from Work after the Surgery

After the surgery, 23 patients did not return to work (47.9% of all those who worked prior to TKA). The reasons for resigning from work, as reported by the participants, are shown in Table 2.

Those who did not return to work were asked whether they would go back to work if certain conditions were met. The majority of the participants would not return to work if they could. The others responded that they would take a job if they could retrain and/or if obstacles in the workplace were removed (see Table 3). 

### 3.4. Assessment of the Participants’ Functional Performance

The mean level of functional performance after TKA assessed using KOS-ADL was 75.89. The mean KOS-ADL score was 76.84 among individuals who were receiving welfare benefits prior to TKA, did not return to work after TKA, and started receiving benefits only after the surgery. The mean KOS-ADL score among individuals who worked prior to TKA and returned to work after TKA was 75.57. This difference was not statistically significant (*p* = 0.055). The lowest level of functional performance was found in the group of individuals doing office work (61.0), while the highest was observed among those doing physical jobs (91.7). The individuals doing physical work in agriculture achieved an average score of 74.0 in KOS-ADL. No statistically significant differences were observed between the groups (*p* = 0.055) (Table 4).

The participants were divided into two groups: those who worked after TKA (regardless of the type of job) and those who did not return to work. The mean KOS-ADL scores in the former and latter groups were 83.67 and 76.67, respectively. This difference was not statistically significant (*p* = 0.55).

The findings show a moderate correlation (r = 4.24; *p* = 0.003) between functional performance in KOS-ADL and self-rated health status. No statistically significant differences in self-rated health status were found between those who did and did not return to work after TKA (*p* = 0.34).

## 4. Discussion

The present study asked three questions, i.e., how many people return to work after TKA one year after surgery, what factors contribute to the decision to resign from work, and what is the level of functional performance among the study participants?

The initial group of potential study participants comprised 431 patients who received unilateral TKA due to knee osteoarthritis. As many as 85.6% of these patients had reached the retirement age by the day of the surgery, which means that nearly 15% of those after TKA were of working age, in line with data in the UK National Joint Registry where those aged 50–59 years in 2020 accounted for 15% of 53,782 total individuals after TKA [34]. This is an important issue due to increasing life expectancy and the need for individuals to continue working in order to maintain quality of life in both material and social terms. However, the present study did not answer an important question, namely, how many of those excluded from the study had discontinued their gainful employment and started receiving welfare benefits due to osteoarthritis before reaching retirement age. This is important in the context of the selected timing of the intervention and the potential return to work. 

In the present study, 31.25% of the participants did not work before the surgery, and they were receiving welfare benefits (a pension) due to a degenerative joint disease of the knee. The group of those who did not work after TKA increased significantly to a level of 79.17%. All non-working participants were receiving a pension because they were unable to perform their previous job and presented a low level of fitness. Only 20.83% of participants, therefore, returned to work, which is a very low rate compared to the 40–98% reported in other studies [43], with the average time to return to work between 8 and 12 weeks after TKA [29,44]. Mongin investigated the RTW rates at one year following TKA or Total Hip Arthroplasty in a group of 241 individuals below 65 years of age. In that group, 144 individuals worked until the day of the surgery. After TKA or THA, an average of 38 patients returned to work (52.7%) after 141 ± 100.69 days, and 34 did not resume work [45].

The present study provides quantitative information on the RTW rates after TKA. To date, very few studies in Poland have reported data related to RTW after hip replacement [46], and no studies have focused on RTW after TKA. A retrospective study by Foote involving 109 individuals under 60 years of age showed that 82% of patients who had worked before surgery and received patellofemoral replacement or unicompartmental replacement returned to work within 12 weeks. Following TKA, 54% of the patients returned to work within 20 weeks after surgery [41]. In the present study, the RTW rate after TKA was 30.3%. Before TKA, 31.25% of the patients were receiving a pension due to osteoarthritis of the knee, whereas after the intervention, 79.16% were receiving a pension due to the disease and their postoperative condition. The low RTW rate in the study group may be due to a number of reasons. Individuals performing physical jobs, including agricultural workers, accounted for 62.5% of the group. Those who did not return to work after TKA noted that their main reason for resigning from work was the inability to perform the job they had before the surgery and the lack of opportunities to retrain or find another job (58.8%). Poor self-assessed health after TKA was the second most important reason for giving up work and was reported by 23.5% of the participants. Only two participants (11.8%) reported that the opportunity to receive welfare benefits was their reason for giving up work, whereas 5.9% of the participants reported a lack of acceptance from their employer as the reason. When asked whether they were informed about the possibility of retraining, the participants who provided answers concerningly reported that they had not been given any such information before or after TKA. Nevertheless, even if certain conditions were met, such as opportunities for retraining or greater employer acceptance, the majority of the participants stated that they would not take a job after TKA. Achieving the highest possible level of functional capacity after TKA is the main goal of therapy. We hypothesized that individuals who returned to work after TKA would present a higher level of functional performance than those who did not return to work after TKA. However, our findings presented a different picture. Those who did not work reported fewer limitations due to pain, swelling, or joint stiffness and fewer limitations related to gait and basic gait function, compared to those who returned to work. Since the difference in functional performance between the two groups was not statistically significant, other factors may have impacted the RTW rate in the study group. No preoperative scores in KOS-ADL are available; therefore, the effects of changes in postoperative functional performance on RTW could not be assessed. However, there are conflicting opinions about the relationship between functional performance and RTW after TKA. Leichtenberg reported a positive relationship between the preoperative KOS-ADL score and RTW after TKA [31], whereas Kuijer found no such relationship between these two factors [47]. In the group of individuals who returned to work, those performing office jobs presented a lower rating of functional performance than that reported by patients who performed physical work in agriculture or other sectors. Consequently, there are no grounds for claiming that physical activity, including rehabilitation, impacted the perception of functional performance in the study group. However, there remains a need for further research to reliably assess the impact of rehabilitation on RTW after TKA.

Currently, there is no consistent process for optimizing RTW among working-age patients after TKA. The effects of late surgical intervention and absent or limited RTW-related counselling contribute to potential delays in returning to work [48,49,50]. Therefore, it is necessary to refocus healthcare provisions for this group of patients, particularly with regard to RTW-oriented rehabilitation. Such provisions could inform patients about the possibility to perform work and contain tailored interventions to optimize patient outcomes [30]. A previous study showed that the process of recovery after TKA might benefit from high-volume exercise applied before surgery. However, assessments of exercise-based interventions applied after surgery did not show the advantage of one exercise program over another [51]. Therefore, there is no clear evidence that rehabilitation after TKA based on a specific type of activity would generally enhance recovery in patients [52,53]. It appears that earlier and more intense physical rehabilitation programs involving exercise (e.g., strength training) produce better effects than less intense programs (e.g., ADL training without strength training) [54]. The most recent reports concerning the impacts of various rehabilitation strategies on RTW outcomes suggest that specific rehabilitation interventions should primarily consider patients’ expectations for their postoperative recovery, their willingness to engage in exercise and physical activity, and their preoperative functional performance [51]. Therefore, future strategies should emphasize the importance of preoperative exercises (pre-habilitation) in patients with reduced preoperative knee extension strength, impaired functional performance, and a tendency to fall [51]. 

In reference to studies carried out to date in various countries, Tilbury noted that the timing of RTW may be affected by the health care and welfare security systems. As an example, Tilbury reported that sick leave from work in the Netherlands is paid in full for the first two years [55]. Lombardi noted that in the USA and other countries, affordable health insurance is an advantage, highlighting a study in which 77% of individuals aged 18–69 years post-TKA obtained benefits from the health insurance provided by their employer [38]. The low RTW rate identified in this study may be linked to the Polish healthcare and social security system. In Poland, welfare benefits are commonly awarded to individuals incapable of working due to a degenerative disease or surgery. Given the relationship between minimum wage and the amount of these benefits, welfare may be an acceptable solution for many people, especially physical workers with low occupational qualifications. Lombardi suggested that one of the main reasons for not returning to work after TKA is retirement and the use of disability benefits prior to surgery [43]. The present findings support this opinion. As many as 31% of the working-age participants in this study received benefits even before the surgery. After TKA, this group increased significantly. 

The presented findings highlighted factors affecting RTW after TKA that are consistent with those reported in other studies. To answer the main question, the group of participants was narrowed to those who were of working age according to Polish regulations and had received surgery within the last year. Unfortunately, the present findings differ significantly from those most frequently reported in other studies. This difference may be due to the small number of participants, but it should be noted that as many as 85% of sampled individuals did not meet the age criterion. The reasons for the low rate of RTW in the study group included a lack of information about the possibility to return to work after surgery, various limitations, and types of work. In Poland, the system of occupational rehabilitation is insufficiently developed. Information on opportunities for retraining or finding another job is not made available to patients with developing osteoarthritis or during the preoperative period. The most common route for the treatment and rehabilitation process is surgery; rehabilitation; satisfactory recovery of functional capacities; and, as demonstrated here, the acquisition of an incapacity pension. This process was confirmed by the findings of our study, which showed that functional performance after TKA was similar between those who returned to work and those that did not.

Considering the growing number of knee replacement surgeries performed on working-age patients, as well as the increasing life expectancy and corresponding need to increase the duration of one’s working life, this group of patients should receive coordinated medical and rehabilitation care. Considering the importance of returning to work among those who have not yet reached retirement age, further large-scale studies should be conducted considering environmental factors and the quality of life among individuals after TKA. It is also necessary to conduct research evaluating the effectiveness of interventions applied in occupational rehabilitation and functional training after TKA. Over the past few years, high hopes have been placed on the modification of surgical technology using computer navigation and robotics. Indeed, new techniques may reduce the number of patients reporting postoperative problems that prevent them from returning to work. Successful RTW is increasingly recognized as an important outcome for this group, with social and economic consequences for patients, employers, and society.

### Study Limitations 

The group of participants was small as a result of the adopted age criterion. It would be necessary to change the age limit to 60 and 65 years in women and men, respectively, or to conduct a multicenter study taking into account different social settings in order to obtain an adequate overview of the problem. The study also did not assess the working hours of those who chose to return to work, so the findings did not indicate whether these participants returned to work full-time or part-time. Another limitation is the fact that no information was acquired on the participants’ rehabilitation process. Indeed, rehabilitation can vary greatly, ranging from the most basic postoperative therapies to in-patient programs at hospital rehabilitation units or systematic treatment programs at rehabilitation centers. Thus, the type of rehabilitation received may be related to one’s decision to return to work. Furthermore, the eligibility criteria in the present study did not include the BMI or height of the patients. One’s BMI and height can considerably impact the recovery of functional capacities after TKA. Thus, these factors should be considered in further research. Future studies should also distinguish between additional sex-related subgroups due to the different age thresholds applied. The division of groups by age and sex is very important, but our sample was narrowed to individuals with primary and unilateral TKA due to degenerative disease, leaving only 48 individuals for the final analysis. For this reason, a further division into subgroups would have made the process of drawing conclusions more difficult. 

## 5. Conclusions

The findings suggest that the rate of RTW after TKA in Poland is significantly lower than that in other countries. The reasons for this situation, as shown in the study, may be related to the lack of an occupational rehabilitation system, resulting in a paucity of information about the possibility to return to work and opportunities for retraining. Patients taking part in this study indicated a need for counselling related to finding work that would match their specific employment requirements. Participants noted that such information could potentially prompt a return to work but also admitted that they did not want to return to work after TKA. The study group mainly included physical workers and agricultural workers, i.e., individuals with limited options for retraining who were also unable to continue to perform physically demanding jobs due to the surgery. It is also necessary to emphasize the importance of the welfare benefits that most of the patients received after TKA since this factor seemed to encourage patients to withdraw from the labor market. 

## Figures and Tables

**Figure 1 jcm-13-01902-f001:**
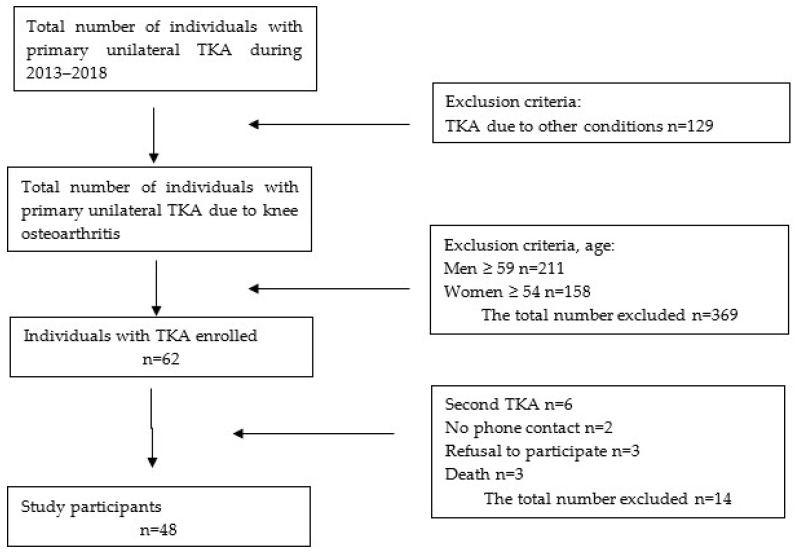
Flow of participants in the study.

**Table 1 jcm-13-01902-t001:** Occupational activities of the participants before and after TKA.

Participants’ Occupational Activities	Before TKA	After TKA
	*N* (%)	*N* (%)
Office work	3 (6.25)	2 (4.17)
Physical work in agriculture	9 (18.75)	2 (4.17)
Other physical jobs	21 (43.75)	6 (12.5)
Welfare benefits	15 (31.25)	38 (79.17)

**Table 2 jcm-13-01902-t002:** The reasons for resigning from work after TKA, as reported by the participants.

Reasons for Resigning from Work	*N*	%
Inability to perform work in the position held before the surgery, with no opportunity to retrain	10	20.8
Poor self-rated health due to arthroplasty and co-existing conditions	4	8.3
The available retirement options	2	4.2
Lack of employer acceptance	1	2.1
No response	31	64.6
Total	48	100.0

**Table 3 jcm-13-01902-t003:** Preconditions for taking a job after TKA.

Would You Go Back to Work after TKA If Certain Conditions Were Met?				
		*N*	Percent	Percentage of Valid Responses
Valid	Opportunity to retrain	2	4.2	11.8
	Removal of obstacles by the employer	2	4.2	11.8
	I would not take a job	15	27.1	76.5
	Total	19	35.4	100.0
No data	No response	29	64.6	
Total		48	100.0	

**Table 4 jcm-13-01902-t004:** Assessment of functional performance after TKA using KOS-ADL.

Occupational Activity after the Surgery	x (sd)	me	*p*
Office work	61.0 (25.46)	61.0	0.055
Physical work in agriculture	74 (9.9)	74.0	
Other physical jobs	91.7 (4.9)	92.0	
Welfare benefits	76.84 (13.6)	80.0	
Total	75.89 (14.02)	80.0	

x, arithmetic mean; sd, standard deviation; me, median.

## Data Availability

The data presented in this study are available upon request from the corresponding author.

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
