# Peer review of "Return to Work after Primary Total Knee Arthroplasty: The First Polish Pilot Retrospective Study"

_jcm, 2024, doi:10.3390/jcm13071902_

Round 1

Reviewer 1 Report

Comments and Suggestions for Authors

Manuscript

1. Include a qualifier to reflect that while TKA is generally effective, outcomes can vary based on several factors - patient health, age, and rehabilitation adherence

2. why is the age limit set at ≤ 54 for women and ≤ 59 for men?

3. authors should acknowledge the use of different prosthetic designs and explain how this was accounted for in the analysis or discuss it as a potential limitation.

4. 48 out of 431 patients met the inclusion criteria. this which raises questions about the representativeness of the sample. Please elaborate on this

5. Provide detail about the statistical analysis - include the rationale for choosing specific tests

6. in the discussion section, authors should discuss potential reasons for the variation in RTW rates compared to other studies

7. consider offering a more detailed analysis or discussion of the relationship between functional capacity (KOS-ADL score) and RTW

8. Discuss potential impacts of various rehabilitation strategies on RTW outcomes

Comments on the Quality of English Language

English

1. "80% majority of the patients" could be "80% of the patients."

2. "Niemcy" should be "Germany"

3. Age of participants is written numerically ("59") and in words ("four")

Author Response

Dear Reviewer,        

We thank you for reviewing our article titled, “Return to work after primary total knee arthroplasty: the first Polish retrospective study”. We have made every effort to improve our manuscript, as guided by the reviewer’s helpful suggestions.

We thank the reviewer of all the comments. Answers are summarised below. All changes are highlighted as red text in the manuscript.

We hope you will be pleased with the changes, and support the publication of our revised manuscript.

With kind regards,

The authors of the article

Point 1: Include a qualifier to reflect that while TKA is generally effective, outcomes can vary based on several factors - patient health, age, and rehabilitation adherence.

Response 1: Thank you for this valuable comment. We fully agree with the Reviewer. In accordance with the Reviewer’s suggestion, we have included a qualifier to reflect that while TKA is generally effective, outcomes can vary based on several factors.

BEFORE

lines 23-29: Introduction

Knee osteoarthritis is a concern for more and more people. It leads to significant impairment of functional capacities, which adversely affects activity and participation, gradually leading to the deterioration of the quality of life [1,2]. Total Knee Arthroplasty (TKA) is the most commonly applied intervention. It is highly effective since it relieves pain and restores the function of the joint, consequently enabling improvement in the patients’ quality of life [3,4].

AFTER

lines 27-49: Introduction

Knee osteoarthritis is a concern for increasingly more people. This affliction leads to significant impairment of functional capacities, which adversely affects activity and participation, gradually leading to a decrease in quality of life [1,2]. Total Knee Arthroplasty (TKA) is the most commonly applied intervention. This treatment is highly effective since it relieves pain and restores the function of the joint, consequently enabling improvements in the patients’ quality of life [3,4]. Although numerous studies have shown that TKA is one of the most successful and effective procedures in orthopedics [5-11], the outcomes can vary based on several factors, such as patient health status, including mental health, as well as expectations, comorbidities, age, preoperative pain ratings, preoperative levels of physical activity, and rehabilitation adherence [12-19]. Additionally, it was suggested that the design of post-TKA rehabilitation should take into account specific patient risk factors, including age, health status, social support status, and knee-related physical functioning before the surgery [20,21]. Furthermore, Styron showed that the factors linked to a faster return to work (RTW) after TKA included female gender, self-employment, better post-operative physical and mental health outcomes, a higher functional comorbidity index, and an accessible workplace accessible. A slower RTW after TKA was linked to the preoperative level of pain, more physically demanding work, and employee compensation [22]. Bardgett identified factors that affected the RTW process after TKA. According to patients, these factors included delays in surgical intervention, limited and often inconsistent RTW-related advice from healthcare professionals, and a lack of rehabilitation to optimize the patient's recovery and facilitate RTW [23].

Point 2: why is the age limit set at ≤ 54 for women and ≤ 59 for men?

Response 2: Thank you for the helpful suggestions. This factor is of key importance in the study. These age criteria for men and women were applied because of the legal retirement age in Poland, women 60, men 65 years. It was assumed that a period of five years preceding the retirement age would be sufficient to investigate RTW rate after the TKA in a working-age population. The explanation has been added in the Methods section.

BEFORE

lines 72-76: Materials and Methods

A retrospective analysis of the medical records was performed, taking into account patients meeting eligibility criteria, i.e., individuals with unilateral TKA, women and men aged  ≤ 54 and ≤ 59 years, respectively, on the day of the surgery (in Poland the retirement age for women and men is 60 and 65 years, respectively), individuals over one year after surgery.

AFTER

lines 107-113: Materials and Methods

A retrospective analysis of the medical records was performed, taking into account patients meeting eligibility criteria, i.e., individuals with unilateral TKA, women aged ≤54 years, and men aged ≤59 years on the day of the surgery. These age criteria for men and women were applied because of the legal retirement age in Poland, which is 60 for women and 65 for men. It was assumed that a period of five years before reaching retirement age and a period of at least one year following TKA would be sufficient for the patients to return to work after TKA.

Point 3: authors should acknowledge the use of different prosthetic designs and explain how this was accounted for in the analysis or discuss it as a potential limitation.

Response 3: Thank you for the helpful suggestions, In accordance with the Reviewer’s suggestion, we have added the explanations, as recommended by the Reviewer.

BEFORE

lines 80-85: Materials and Methods

Cruciate retaining prostheses of two designs were applied, i.e. Triathlon (Stryker Orthopaedics, Mahwah, New Jersey), and PFC Sigma (Depuy Synthes, Raynham, Massachusetts). The patella was not routinely resurfaced.

The procedure applied primary condylar endoprosthesis Columbus (Aesculap BBraun, Tuttlingen, Niemcy) and Vanguard (Biomet, Warsaw, USA) with similar kinematic characteristics. Patellar endoprosthesis was not applied.

AFTER

lines 118-129: Materials and Methods

Because of their similar construction design, kinematics, instrumentation, and corresponding surgical techniques, prostheses from two companies were selected: Columbus™ (Aesculap B Braun, Tuttlingen, Germany) (n=30) and Vanguard (Biomet, Warsaw, USA) (n=18). Both designs are primary, condylar prostheses with over 15 years of orthopedic application and include unconstrained posterior cruciate retaining (CR). In addition, intact PCL provides stability under flexion and allows for femoral rollback.

Polyethylene–metal bearings were applied in all cases, and the same approach was used, i.e., the medial parapatellar approach with classic instrumentation, extracortical fixation for tibia first, and intramedullary femoral instrumentation. The same surgical technique for insertion and theory of component fixation were also applied. All procedures were preceded by preoperative planning in the mediCAD software (mediCAD Hectec GmbH, Landshut, Germany). No patelloplasty was performed.

Point 4: 48 out of 431 patients met the inclusion criteria. this which raises questions about the representativeness of the sample. Please elaborate on this.

Response 4: Thank you for the helpful suggestions. The minimum sample size was calculated relative to the total number of TKA surgeries administered in 2021 in the Podkarpackie Region, Poland. Prior to the recruitment procedure (retrospective analysis of medical records), analysis of power was computed using Statistica, version 13.1 (StatSoft Poland). The general indicator of effect size for all outcome measures and the power of the study were 0.5, the probability was 0.05 and the maximum error was 10%. The estimated target sample size was 45, as a result 48 participants were enrolled for this study.

The detailed flow of participants in the study is shown in Figure 1. The information was added in the Materials and Results sections.

BEFORE

lines 112-119: Materials and Methods

Calculations were performed using the IBM SPSS programme. For nominal and ordinal variables, results were presented as relative and absolute numbers (% and N). Results for quantitative variables were presented using descriptive statistics (arithmetic mean, median, standard deviation). Significance of the differences in the quantitative variables between groups was assessed using Mann-Whitney test. Assessment of the differences in the nominal variables between the groups was performed using chi-squared test. The Spearman correlation coefficient was used to assess the relationship; the limit of statistical significance in the tests performed was defined as p<0.05.

lines 122-126: Results

Based on medical records 431 patients were selected for the study. The age criterion was exceeded by 85.6% of the patients (369 individuals). Following the first contact by phone, 14 individuals were excluded (3 individuals were dead, 3 patients refused to participate, 2 individuals changed their phone number, 6 patients reported they had another TKA).

AFTER:

lines 164-77: Materials and Methods

Prior to the recruitment procedure (a retrospective analysis of medical records), the analysis of power was computed using Statistica, version 13.1 (StatSoft Poland). The general indicator of effect size for all outcome measures and the power of the study were 0.5, the probability was 0.05, and the maximum error was 10%. The estimated target sample size was 45. Ultimately, 48 participants were enrolled in this study.

Statistical analyses were computed using SPSS (IBM, Chicago, IL, USA), version 21.0. Categorical variables were reported as percentage values, and continuous variables were presented as the mean, standard deviation, and median values. Analysis of the normal distribution was carried out using a Shapiro–Wilk test. Comparative analysis of the KOS-ADL scores after surgery in the occupational activity groups was performed using the non-parametric Mann–Whitney test. The Spearman correlation coefficient was used to assess the relationship between functional performance measured with KOS-ADL and self-rated health status. Statistical significance was confirmed at p < 0.05.

lines 180-185: Results

Based on medical records, 431 patients were selected for this study. The age criterion was exceeded by 85.6% of the patients (369 individuals). Following the first discussion by phone, fourteen individuals were excluded (three individuals had died, three patients refused to participate, two individuals changed their phone number, and six patients reported that they had another TKA). The detailed flow of participants in the study is shown in Figure 1.

Point 5: Provide detail about the statistical analysis - include the rationale for choosing specific tests.

Response 5: Thank you for the helpful suggestions, In accordance with the Reviewer’s suggestion, information about the statistical methods applied has been improved. Statistical analyses were computed using SPSS (IBM, Chicago, IL, USA), version 21.0. Categorical variables were reported as percent values, and continuous variables were shown as mean, standard deviation, and median values. Analysis of normal distribution was carried out using the Shapiro–Wilk test. Comparative analysis of the KOS-ADL scores after the surgery in the occupational activity groups was performed using the non-parametric Mann-Whitney test. It was assumed that statistical significance was reflected by p < 0.05. The information has been added in the Statistical Methods section.

BEFORE

lines 112-119: Statistical methods

Calculations were performed using the IBM SPSS programme. For nominal and ordinal variables, results were presented as relative and absolute numbers (% and N). Results for quantitative variables were presented using descriptive statistics (arithmetic mean, median, standard deviation). Significance of the differences in the quantitative variables between groups was assessed using Mann-Whitney test. Assessment of the differences in the nominal variables between the groups was performed using chi-squared test. The Spearman correlation coefficient was used to assess the relationship; the limit of statistical significance in the tests performed was defined as p<0.05.

AFTER:

lines 164-77: Statistical methods

Prior to the recruitment procedure (a retrospective analysis of medical records), the analysis of power was computed using Statistica, version 13.1 (StatSoft Poland). The general indicator of effect size for all outcome measures and the power of the study were 0.5, the probability was 0.05, and the maximum error was 10%. The estimated target sample size was 45. Ultimately, 48 participants were enrolled in this study.

Statistical analyses were computed using SPSS (IBM, Chicago, IL, USA), version 21.0. Categorical variables were reported as percentage values, and continuous variables were presented as the mean, standard deviation, and median values. Analysis of the normal distribution was carried out using a Shapiro–Wilk test. Comparative analysis of the KOS-ADL scores after surgery in the occupational activity groups was performed using the non-parametric Mann–Whitney test. The Spearman correlation coefficient was used to assess the relationship between functional performance measured with KOS-ADL and self-rated health status. Statistical significance was confirmed at p < 0.05.

Point 6: in the discussion section, authors should discuss potential reasons for the variation in RTW rates compared to other studies.

Response 6: Thank you for the helpful suggestions, In accordance with the Reviewer’s suggestion, we have discussed potential reasons for the variation in RTW rates compared to other studies in the discussion section. These have been highlighted in the Discussion.

The current findings identify factors affecting RTW after TKA, that are consistent with those reported in other studies. In order to answer the main question, the group of participants was narrowed down to those of working age according to Polish regulations, and one year after the surgery. Unfortunately, the present findings differ significantly from those most frequently reported in other studies. This may be due to the small number of participants, but it should be remembered that as many as 85% of the individuals did not meet the age criterion. The reasons for the low rate of RTW in the study group include lack of information about the possibility to go back to work after the surgery, as well as limitations and types of work. In Poland the system of occupational rehabilitation is insufficiently developed. Information about opportunities for retraining or taking up another job, is not available when osteoarthritis is developing or during the pre-operative period. The commonly accepted route for the treatment and rehabilitation process is surgery, rehabilitation, and satisfactory recovery of functional capacities and, finally, as demonstrated here, acquisition of the incapacity pension. This is confirmed by the findings of our study showing that functional performance after TKA is similar in both the individuals who returned to work and those that did not.

BEFORE

lines 291-299: Discussion

Tilbury points out, in reference to the studies carried out to date in various countries, that it cannot be ruled out that the timing of RTW may be affected by the health care system or welfare security system. As an example he reports that the sick leave from work in the Netherlands is paid in full for the first two years [27]. Lombardi points out that in the USA and other countries, affordable health insurance is an advantage and refers to a study in which 77% of the individuals aged 18-69 years post-TKA obtained benefits from health insurance provided by their employer [19]. Perhaps the low RTW rate identified in this study may be linked to the Polish healthcare and social security system.

AFTER:

lines 377-409: Discussion

In reference to studies carried out to date in various countries, Tilbury noted that the timing of RTW may be affected by the health care and welfare security systems. As an example, Tilbury reported that sick leave from work in the Netherlands is paid in full for the first two years [49]. Lombardi noted that in the USA and other countries, affordable health insurance is an advantage, highlighting a study in which 77% of individuals aged 18-69 years post-TKA obtained benefits from the health insurance provided by their employer [38]. The low RTW rate identified in this study may be linked to the Polish healthcare and social security system. In Poland, welfare benefits are commonly awarded to individuals incapable of working due to a degenerative disease or surgery. Given the relationship between minimum wage and the amount of these benefits, welfare may be an acceptable solution for many people, especially physical workers with low occupational qualifications. Lombardi suggested that one of the main reasons for not returning to work after TKA is retirement and the use of disability benefits prior to surgery [38]. The present findings support this opinion. As many as 31% of the working-age participants in this study received benefits even before the surgery. After TKA, this group increased significantly.

The presented findings highlighted factors affecting RTW after TKA that are consistent with those reported in other studies. To answer the main question, the group of participants was narrowed to those who were of working age according to Polish regulations and had received surgery within the last year. Unfortunately, the present findings differ significantly from those most frequently reported in other studies. This difference may be due to the small number of participants, but it should be noted that as many as 85% of sampled individuals did not meet the age criterion. The reasons for the low rate of RTW in the study group included a lack of information about the possibility to return to work after surgery, various limitations, and types of work. In Poland, the system of occupational rehabilitation is insufficiently developed. Information on opportunities for retraining or finding another job is not made available to patients with developing osteoarthritis or during the pre-operative period. The most common route for the treatment and rehabilitation process is surgery; rehabilitation; satisfactory recovery of functional capacities; and, as demonstrated here, the acquisition of an incapacity pension. This process was confirmed by the findings of our study, which showed that functional performance after TKA was similar between those who returned to work and those that did not.

Point 7: consider offering a more detailed analysis or discussion of the relationship between functional capacity (KOS-ADL score) and RTW.

Response 7: Thank you for the helpful suggestions. In accordance with the Reviewer’s suggestion, we have added more detailed discussion of the relationship between functional capacity (KOS-ADL score) and RTW.

BEFORE

lines 266-274: Discussion

It is interesting to analyse the results of the participants’ functional assessment and to compare the KOS-ADL score of those who returned to work with those who did not. The difference was not statistically significant, so the decision about RTW was impacted not only by the fitness level but by other factors, probably social. There are conflicting opinions about the relationship between functional capacity and RTW after TKA. Leichtenberg assessed preoperative functional capacity with the KOS-ADL scale and found that there was a relationship between KOS-ADL score and RTW after TKA [10] In contrast, Kuijer did not observe a significant relationship between KOS-ADL score and RTW [26].

AFTER:

lines 324-345: Discussion

Achieving the highest possible level of functional capacity after TKA is the main goal of therapy. We hypothesized that individuals who returned to work after TKA would present a higher level of functional performance than those who did not return to work after TKA. However, our findings presented a different picture. Those who did not work reported fewer limitations due to pain, swelling, and joint stiffness, as well as fewer limitations related to gait and basic gait function, compared to those who returned to work. Overall, the difference in functional performance between the two groups was not statistically significant. Therefore, other factors may have impacted the decision to return to or resign from work among the study group. Since no assessment with KOS-ADL was performed before TKA, it was impossible to assess the effects of changes in functional performance on RTW after surgery. However, there are conflicting opinions about the relationship between functional performance and RTW after TKA. Leichtenberg assessed pre-operative functional performance using KOS-ADL and reported a positive relationship between the KOS-ADL score and RTW after TKA [29]. Conversely, Kuijer did not find a significant relationship between the KOS-ADL score and RTW [41]. In the group of individuals who returned to work, those performing office jobs presented a lower rating of functional performance than that reported by patients who performed physical work in agriculture or other sectors. Consequently, there are no grounds for claiming that physical activity, including rehabilitation, impacted the perception of functional performance in the study group. However, there remains a need for further research to reliably assess the impact of rehabilitation on RTW after TKA.

Point 8: Discuss potential impacts of various rehabilitation strategies on RTW outcomes.

Response 8: Thank you for the helpful suggestions. In accordance with the Reviewer’s suggestion, we have discussed potential impacts of various rehabilitation strategies on RTW outcomes.

BEFORE

lines 275-280: Discussion

Currently, there is no consistent process for optimising RTW of working-age patients after TKA. The effects of late surgical intervention, as well as a lack of or limited RTW-related counselling contribute to potential delays in return to work [23,24,25]. Therefore, it is necessary to refocus healthcare provision for this group of patients, particularly with regard to RTW-oriented rehabilitation informing about the possibility to per-form work and containing tailored interventions to optimise patient outcomes [22].

AFTER:

lines 346-366: Discussion

Currently, there is no consistent process for optimizing RTW among working-age patients after TKA. The effects of late surgical intervention and absent or limited RTW-related counselling contribute to potential delays in return to work [42-44]. Therefore, it is necessary to refocus healthcare provisions for this group of patients, particularly with regard to RTW-oriented rehabilitation. Such provisions could inform patients about the possibility to perform work and contain tailored interventions to optimize patient outcomes [23]. A previous study showed that the process of recovery after TKA might benefit from high-volume exercise applied before surgery. However, assessments of exercise-based interventions applied after surgery did not show the advantage of one exercise program over another [45]. Therefore, there is no clear evidence that rehabilitation after TKA based on a specific type of activity would generally enhance recovery in patients [46,47]. It appears that earlier and more intense physical rehabilitation programs involving exercise (e.g., strength training) produce better effects than less intense programs (e.g., ADL training without strength training) [48]. The most recent reports concerning the impacts of various rehabilitation strategies on RTW outcomes suggest that specific rehabilitation interventions should primarily consider patients' expectations for their post-operative recovery, their willingness to engage in exercise and physical activity, and their pre-operative functional performance [45]. Therefore, future strategies should emphasize the importance of pre-operative exercises (pre-habilitation) in patients with reduced pre-operative knee extension strength, impaired functional performance, and a tendency to fall [45].

Point 9: English

  1. "80% majority of the patients" could be "80% of the patients."

  1. "Niemcy" should be "Germany"

  1. Age of participants is written numerically ("59") and in words ("four")

Response 9: Thank you for the helpful suggestions. In accordance with the suggestion, the manuscript has been proofread by MDPI English Language Editing Services and we have received the certificate of language editing use.

Reviewer 2 Report

Comments and Suggestions for Authors

Even if this is an interesting study (retrospective cohort) on return to work after total knee arthroplasty, methodology and design need to be enhanced. Final recruitment group (taking account inclusion and exclusion criteria) is restricted, when compared with total number of TKAs. English used are readable and but minor editing is required. Several points should be revised in order to improve this paper.  These points include:

Title:  a brief and comprehensive title, which could be completed according to the comments in order to be more specific.

Abstract: a comprehensive presentation of the interesting aspects of the study.

Introduction:  The extension of this session is acceptable. However, some interesting topics could be enhanced with additional literature according to the comments, in order to better explain the results of this study. Marked sentences should be completed or rephrased.

Methodology: Some important points of this session could be changed. Study design could be better presented and marked sentences further analyzed. Exclusion and inclusion criteria could be more specific regarding some important topics like measurements including BMI and height of the patients. The way self-assessment of health was evaluated should be defined (whether an already validated scale has been used). When, it is feasible, some additional subgroups could be presented (related with sex, due to different age threshold applied).

Results: This session could be minimized and information already presented in the tables could be removed from the text, which should be focused on the most important, ones. Results could be divided in subgroups, taking account different age threshold. Marked points could be more clearly presented.  

Discussion: An extended presentation of the literature regarding the points under investigation. Some paragraphs of this session could be incorporated in the introduction session and some others could be removed in order to enhance conclusions session.  Paragraphs analyzing known literature could be shortened and these used to explain the results of this study could be presented with tables, summarizing results. An interesting comparison of KOOS-ADL with known literature is applied , but the comparison with the pre-operative values could be useful, if feasible.

Moreover, there are existing studies in the literature on TKA that investigate the correlations between KOOS and other patient-reported outcomes with various parameters. Exploring these papers could be beneficial for your discussion section.

Conclusions: Conclusions should be focused on what the results of this study can offer to the clinicians regarding the results of TKAs and especially return to work. This session could be further enhanced with comments from the discussion, too.

Comments on the Quality of English Language

moderate english editing is needed

Author Response

Dear Reviewer,        

We thank you for reviewing our article titled, “Return to work after primary total knee arthroplasty: the first Polish retrospective study”. We have made every effort to improve our manuscript, as guided by the reviewer’s helpful suggestions.

We thank the reviewer of all the comments. Answers are summarised below. All changes are highlighted as red text in the manuscript.

We hope you will be pleased with the changes, and support the publication of our revised manuscript.

With kind regards,

The authors of the article

Point 1: Even if this is an interesting study (retrospective cohort) on return to work after total knee arthroplasty, methodology and design need to be enhanced. Final recruitment group (taking account inclusion and exclusion criteria) is restricted, when compared with total number of TKAs. English used are readable and but minor editing is required.

Response 1: Thank you for this valuable comment. All the suggestions have been addressed in the revised manuscript. We fully agree with the Reviewer. In accordance with the Reviewer’s suggestion, we have enhanced methodology and design. The aim of the study has been rewritten to make it more specific and accurate. We have described the group of participants as well as the inclusion and exclusion criteria in more detail. We have added a figure presenting the flow of participants in the study (in the Results section). The minimum sample size was calculated relative to the total number of TKA surgeries administered in 2021 in the Podkarpackie Region, Poland. Prior to the recruitment procedure (retrospective analysis of medical records), analysis of power was computed using Statistica, version 13.1 (StatSoft Poland). The general indicator of effect size for all outcome measures and the power of the study were 0.5, the probability was 0.05 and the maximum error was 10%. The estimated target sample size was 45, as a result 48 participants were enrolled for this study.

The manuscript has been proofread by MDPI English Language Editing Services and we have received the certificate of language editing use.

BEFORE

lines 62-64: Introduction

To our knowledge, no studies so far have investigated the level of participation and RTW after TKA in Poland. Hence, this study aimed to identify predictors of RTW after TKA in patients of working age (≤ 65 years) in Poland.

AFTER

lines 94-99: Introduction

However, there are no reliable data on Polish patients. For this reason, the present retrospective study was designed to determine the RTW rates after TKA, provide reference data, and identify the factors impacting patients’ decisions to return to or resign from work, as well as their functional capacities following TKA. It was hypothesized that RTW rates in patients after TKA in Poland would be similar to those in developed countries.

BEFORE

lines 62-64: Materials and Methods

2.1. Participants

The study was conducted in a group of patients treated at the Department of Orthopaedics and Traumatology of the Regional Clinical Hospital No. 2 in Rzeszów, Poland between 2013 and 2018. All surgical procedures were performed by the same team of five orthopaedists, using the same surgical technique: tibia first and mechanical alignment (TF&MA).

A retrospective analysis of the medical records was performed, taking into account patients meeting eligibility criteria, i.e., individuals with unilateral TKA, women and men aged  ≤ 54 and ≤ 59 years, respectively, on the day of the surgery (in Poland the retirement age for women and men is 60 and 65 years, respectively), individuals over one year after surgery. Exclusion criteria: revision knee arthroplasty and simultaneous bilateral TKA, other musculoskeletal diseases (e.g. spinal stenosis), a history of other surgeries or internal diseases making it impossible to return to work, oncological diseases, musculoskeletal infections and lack of consent to participate.

Cruciate retaining prostheses of two designs were applied, i.e. Triathlon (Stryker Or-thopaedics, Mahwah, New Jersey), and PFC Sigma (Depuy Synthes, Raynham, Massa-chusetts). The patella was not routinely resurfaced.

The procedure applied primary condylar endoprosthesis Columbus (Aesculap BBraun, Tuttlingen, Niemcy) and Vanguard (Biomet, Warsaw, USA) with similar kinematic characteristics. Patellar endoprosthesis was not applied.

In the total group of 431 patients after TKA, inclusion criteria were met by 48 patients, 23 females and 25 males. On average the participants were aged 59 (±4.09) years.

All the patients gave their consent to participate in the study. The study was ap-proved by the Bioethics Commission of the Centre for Post-graduate Medical Training in Warsaw (No. 71/PB/2012).

2.2. Measures

 All the patients selected based on a review of medical records were approached by phone by a doctor, member of the hospital staff and the research team. A phone interview, based on a questionnaire, was conducted with all participants by a member of the team. The 14-item questionnaire was designed by the team of researchers (orthopaedists and experienced physiotherapists). The first part of the questionnaire was related to the demographics and body size measures. The subsequent items asked about the date of surgery, post-operative rehabilitation and post-operative complications. The final part focused on the participants’ occupational status before and after TKA, factors that influenced the decision to return to work and self-assessment of health.

Functional limitations in everyday life due to a knee dysfunction following TKA were assessed using the Knee Outcome Survey - Activities of Daily Living (KOS-ADL) [12]. This is a reliable and accurate instrument that responds to changes in the functional performance of patients with various knee conditions who receive physiotherapy or orthopaedic interventions  [13,14]. The study applied a Polish version of KOS-ADL [15]. Responses are given on a scale of 0-5. The maximum final score is 70 points. To calculate the Activity in Daily Living (ADL) index, the final score is divided by 70 and multiplied by 100. KOS-ADL scores are presented for values normalised to the range 0-100.

2.3. Statistical methods

Calculations were performed using the IBM SPSS programme. For nominal and ordinal variables, results were presented as relative and absolute numbers (% and N). Results for quantitative variables were presented using descriptive statistics (arithmetic mean, median, standard deviation). Significance of the differences in the quantitative variables between groups was assessed using Mann-Whitney test. Assessment of the differences in the nominal variables between the groups was performed using chi-squared test. The Spearman correlation coefficient was used to assess the relationship; the limit of statistical significance in the tests performed was defined as p<0.05.

AFTER

lines 100-177: Materials and Methods

2.1. Participants

This retrospective, single-center study was conducted on a group of patients treated at the Department of Orthopedics and Traumatology of the Regional Clinical Hospital No. 2 in Rzeszów, Poland, between 2013 and 2018. All surgical procedures were performed by the same team of five orthopedists using the same surgical technique: tibia first and mechanical alignment (TF&MA).

A retrospective analysis of the medical records was performed, taking into account patients meeting eligibility criteria, i.e., individuals with unilateral TKA, women aged ≤54 years, and men aged ≤59 years on the day of the surgery. These age criteria for men and women were applied because of the legal retirement age in Poland, which is 60 for women and 65 for men. It was assumed that a period of five years before reaching retirement age and a period of at least one year following TKA would be sufficient for the patients to return to work after TKA. Exclusion criteria included revision knee arthroplasty and simultaneous bilateral TKA, other musculoskeletal diseases (e.g., spinal stenosis), a history of other surgeries or internal diseases making it impossible to return to work, oncological diseases, musculoskeletal infections, and lack of consent to participate.

Because of their similar construction design, kinematics, instrumentation, and corresponding surgical techniques, prostheses from two companies were selected: Columbus™ (Aesculap B Braun, Tuttlingen, Germany) (n=30) and Vanguard (Biomet, Warsaw, USA) (n=18). Both designs are primary, condylar prostheses with over 15 years of orthopedic application and include unconstrained posterior cruciate retaining (CR). In addition, intact PCL provides stability under flexion and allows for femoral rollback.

Polyethylene–metal bearings were applied in all cases, and the same approach was used, i.e., the medial parapatellar approach with classic instrumentation, extracortical fixation for tibia first, and intramedullary femoral instrumentation. The same surgical technique for insertion and theory of component fixation were also applied. All procedures were preceded by preoperative planning in the mediCAD software (mediCAD Hectec GmbH, Landshut, Germany). No patelloplasty was performed.

In the total group of 431 patients after TKA, the inclusion criteria were met by 48 patients, including 23 females and 25 males. On average, the participants were aged 59 (±4.09) years.

All patients gave their consent to participate in the study. The study was approved by the Bioethics Commission of the Center for Post-graduate Medical Training in Warsaw (No. 71/PB/2012).

2.2. Measures

 All patients were selected based on a review of medical records. These patients were approached via phone by a doctor, a member of the hospital staff, and the research team. A phone interview based on a questionnaire was conducted with all participants and led by a member of the team. The 14-item questionnaire was designed by the team of researchers (orthopedists and experienced physiotherapists). The first part of the questionnaire was related to the demographics and body size measurements. The subsequent items asked about the date of surgery, post-operative rehabilitation, and post-operative complications. The final part focused on the participants’ occupational status before and after TKA, factors that influenced the decision to return to work and the self-assessment of health.

The patients were asked to assess their health status on a 5-point Likert-type scale by responding to the following question: “Which term most accurately describes your current health compared to others in the same age group?” Possible responses were as follows: very good (=1), good (=2), moderate (=3), not good (=4), or poor (=5). The scoring system was inverted to make the results easier to read (higher scores indicated better self-rated health) [32].

Functional limitations in everyday life due to a knee dysfunction following TKA were assessed using the Knee Outcome Survey—Activities of Daily Living (KOS-ADL) [33]. This survey is a reliable and accurate instrument that responds to changes in the functional performance of patients with various knee conditions who receive physiotherapy or orthopedic interventions [34,35]. The study applied a Polish version of KOS-ADL [36], and responses were given on a scale of 0-5. The maximum final score was 70 points. To calculate the Activity in Daily Living (ADL) index, the final score was divided by 70 and multiplied by 100. KOS-ADL scores were presented for values normalized to a range of 0-100.

2.3. Statistical methods

Prior to the recruitment procedure (a retrospective analysis of medical records), the analysis of power was computed using Statistica, version 13.1 (StatSoft Poland). The general indicator of effect size for all outcome measures and the power of the study were 0.5, the probability was 0.05, and the maximum error was 10%. The estimated target sample size was 45. Ultimately, 48 participants were enrolled in this study.

Statistical analyses were computed using SPSS (IBM, Chicago, IL, USA), version 21.0. Categorical variables were reported as percentage values, and continuous variables were presented as the mean, standard deviation, and median values. Analysis of the normal distribution was carried out using a Shapiro–Wilk test. Comparative analysis of the KOS-ADL scores after surgery in the occupational activity groups was performed using the non-parametric Mann–Whitney test. The Spearman correlation coefficient was used to assess the relationship between functional performance measured with KOS-ADL and self-rated health status. Statistical significance was confirmed at p < 0.05.

BEFORE

lines 120-126: Results

3.1. The flow of participants

Based on medical records 431 patients were selected for the study. The age criterion was exceeded by 85.6% of the patients (369 individuals). Following the first contact by phone, 14 individuals were excluded (3 individuals were dead, 3 patients refused to participate, 2 individuals changed their phone number, 6 patients reported they had another TKA).

AFTER

lines 180-185: Results

3.1. The flow of participants

Based on medical records, 431 patients were selected for this study. The age criterion was exceeded by 85.6% of the patients (369 individuals). Following the first discussion by phone, fourteen individuals were excluded (three individuals had died, three patients refused to participate, two individuals changed their phone number, and six patients reported that they had another TKA). The detailed flow of participants in the study is shown in Figure 1.

Point 2: Several points should be revised in order to improve this paper. These points include:

Title:  a brief and comprehensive title, which could be completed according to the comments in order to be more specific.

Response 2: Thank you for the helpful suggestions. In accordance with the Reviewer’s comment we have changed the title to make it more specific.

BEFORE

line 2: Return to work after knee arthroplasty: a cohort study

AFTER

line 2: Return to work after primary total knee arthroplasty: the first Polish retrospective study

Point 3: Abstract: a comprehensive presentation of the interesting aspects of the study.

Response 3: Thank you for the helpful suggestions. We have revised the Abstract accordingly, and in line with the instructions of the Journal so that it does not exceed 200 words.

BEFORE

lines 9-20: Abstract

1)Background: No studies so far have investigated the level of participation and return to work (RTW) after total knee arthroplasty (TKA) in Poland. The present study aimed to gain insight into RTW outcomes following TKA in patients of working age (≤ 65 years) in Poland. 2)Methods: The retrospective study was conducted in a group of 431 patients. Ultimately, 48 individuals were qualified. An interview was carried out regarding RTW, and the factors impacting the decision to return to or resign from work. Functional performance was assessed using the Knee Outcome Survey Activities of Daily Living (KOS-ADL). 3)Results: Before TKA, 31.25 % of those qualified for the study did not work, and they were receiving welfare benefits. After the surgery 23 individuals did not return to work (47.9% of all those working prior to TKA). The group of those who did not work after TKA increased significantly, to a level of 79.17%. 4)Conclusion: Bearing in mind how important it is to return to work for people who have not reached the retirement age, further large-scale studies should be conducted taking into account environmental factors and the quality of life in individuals after TKA.

AFTER

lines 9-22: Abstract

1) Background: Total knee arthroplasty (TKA) performed on working-age patients significantly affects the participation of such patients in social life. A retrospective study was conducted to determine the return to work (RTW) rate after TKA. The goal of this study was to provide reference data for the Polish population and identify the factors impacting patients’ decisions to return to or resign from work, relative to their functional performance. 2) Methods: This retrospective study analyzed 431 patients. Ultimately, 48 individuals were enrolled. For this study, an interview on RTW was carried out. This interview also explored the factors impacting a patient’s decision to return to or resign from work. Functional performance was assessed using the Knee Outcome Survey Activities of Daily Living (KOS-ADL). 3) Results: Before TKA, 31.25% of those qualified for the study did not work and were receiving welfare benefits. After the surgery, 23 individuals did not return to work (47.9% of all those working prior to TKA). The group of those who did not work after TKA increased significantly, reaching 79.17%. 4) Conclusions: Considering the importance of returning to work for those who have not reached retirement age, further large-scale studies should be conducted while taking into account environmental factors and the quality of life among individuals after TKA.

Point 4: Introduction:  The extension of this session is acceptable. However, some interesting topics could be enhanced with additional literature according to the comments, in order to better explain the results of this study. Marked sentences should be completed or rephrased.

Response 4: Thank you for the helpful suggestions. In accordance with the Reviewer’s recommendation, we have extended the Introduction by covering some other issues based on newly added references, in order to better explain the results of this study.

BEFORE

lines 24-61: Introduction

Knee osteoarthritis is a concern for more and more people. It leads to significant impairment of functional capacities, which adversely affects activity and participation, gradually leading to the deterioration of the quality of life [1,2]. Total Knee Arthroplasty (TKA) is the most commonly applied intervention. It is highly effective since it relieves pain and restores the function of the joint, consequently enabling improvement in the patients’ quality of life [3,4].

In Poland, according to the data from the National Health Fund, as many as 37,821 knee replacement procedures were performed in 2020, including 30,615 TKA procedures. Primary knee arthroplasty was performed in 94.3% of the cases, and 95% of the patients were operated due to primary gonarthrosis. Women accounted for 72% of all the patients who received this treatment, and 80% majority of the patients were aged 60-79 years. On average the female and male patients were aged 69 and 67 years, respectively. One in two patients who received knee replacement in 2019 were up to 69 years of age [5]. According to the data published by The United Kingdom National Joint Registry, in 2016 patients up to 65 years of age accounted for 40% of the cases [6]. The number of patients with diagnosed knee osteoarthritis is consistently growing. Similarly, the number of TKA procedures performed annually is increasing, and individuals up to 65 years of age are expected to be the largest group among those receiving the treatment [7].

In view of the fact that the duration of working life is increasing, return to work (RTW) after TKA is an issue of great importance. Due to the scarcity of the related research, the determinants of RTW after TKA are still unclear. It is difficult to compare and analyse the results because of the widely varied definitions of work status and different time frames used to measure it. In 2014, Tilbury presented the findings of a systematic review of studies investigating RTW after hip or knee replacement [8]. It showed that the rates of RTW after TKA ranged from 71 to 83% at 3-6 months. Furthermore, Van Leemput reported that the percentage of individuals working after TKA was in the range between 36% and 89%, and those who worked before and returned to work after TKA accounted for 40 to 98% of the populations studied, whereas the mean duration of the rehabilitation period after TKA ranged from 7.7 to 16.6 weeks [8]. According to Van Leemput, slower or no RTW after TKA was more likely to be observed in patients performing jobs of a more physical nature, or those who were absent from work before the surgery, however, most patients subjected to TKA returned to work after the procedure [9]. Leichtenberg established that more advanced age, lower educational and preoperative absence from work were associated with partial or no return to work [10]. A study by Scott took into account 289 patients after TKA. Multivariate analysis showed that heavy or moderate physical work as well as age were the main factors contributing to a return to any employment or return to the same job. In the group of patients working before TKA, all those below 50 years of age and 60% of those aged 50-54 years returned to any type of work [11].

AFTER

lines 27-93: Introduction

Knee osteoarthritis is a concern for increasingly more people. This affliction leads to significant impairment of functional capacities, which adversely affects activity and participation, gradually leading to a decrease in quality of life [1,2]. Total Knee Arthroplasty (TKA) is the most commonly applied intervention. This treatment is highly effective since it relieves pain and restores the function of the joint, consequently enabling improvements in the patients’ quality of life [3,4]. Although numerous studies have shown that TKA is one of the most successful and effective procedures in orthopedics [5-11], the outcomes can vary based on several factors, such as patient health status, including mental health, as well as expectations, comorbidities, age, preoperative pain ratings, preoperative levels of physical activity, and rehabilitation adherence [12-19]. Additionally, it was suggested that the design of post-TKA rehabilitation should take into account specific patient risk factors, including age, health status, social support status, and knee-related physical functioning before the surgery [20,21]. Furthermore, Styron showed that the factors linked to a faster return to work (RTW) after TKA included female gender, self-employment, better post-operative physical and mental health outcomes, a higher functional comorbidity index, and an accessible workplace accessible. A slower RTW after TKA was linked to the preoperative level of pain, more physically demanding work, and employee compensation [22]. Bardgett identified factors that affected the RTW process after TKA. According to patients, these factors included delays in surgical intervention, limited and often inconsistent RTW-related advice from healthcare professionals, and a lack of rehabilitation to optimize the patient's recovery and facilitate RTW [23].

According to the data from the National Health Fund, in Poland, as many as 37,821 knee replacement procedures were performed in 2020, including 30,615 TKA procedures. Primary knee arthroplasty was performed in 94.3% of the cases, and 95% of the patients were operated on due to primary gonarthrosis. Women accounted for 72% of all patients who received this treatment, and 80% of the patients were aged 60-79 years. On average, the female and male patients were aged 69 and 67 years, respectively. One in two patients who received a knee replacement in 2019 were 69 years of age or younger [24]. According to data published by The United Kingdom National Joint Registry, in 2016, patients up to 65 years of age accounted for 40% of cases [25]. Notably, the number of patients with diagnosed knee osteoarthritis is consistently growing. Similarly, the number of TKA procedures performed annually is increasing, and individuals up to 65 years of age are expected to be the largest group among those receiving this treatment [26].

Considering that the duration of people’s working lives is increasing, return to work (RTW) after TKA is an issue of great importance. Due to the scarcity of related research, the determinants of RTW after TKA are still unclear. It also remains difficult to compare and analyze the results because the definitions of work status vary widely, and different timeframes are often used to measure this status. In 2014, Tilbury presented the findings of a systematic review of studies investigating RTW after hip or knee replacement [27]. This review showed that the percentage of patients returning to work was between 25% and 95% in the period from 1 to 12 months after THA and between 71% and 83% from 3 to 6 months after TKA [27]. Furthermore, Van Leemput reported that the percentage of individuals working after TKA was between 36% and 89%. Those who worked before and returned to work after TKA accounted for 40% to 98% of the populations studied, whereas the mean duration of the rehabilitation period after TKA ranged from 7.7 to 16.6 weeks [28]. According to Van Leemput, a slower or no RTW after TKA was more likely to be observed in patients performing jobs of a more physical nature or among those who were absent from work before the surgery. However, most patients subjected to TKA returned to work after the procedure [28]. Leichtenberg established that a more advanced age, lower educational level, and preoperative absence from work were associated with a partial or no return to work [29]. More specifically, Leichtenberg reported that in a post-TKA group of 56 individuals aged below 65 years (mean 56 years), there were 40 individuals (71%) who worked prior to the surgery and returned to full-time work one year after the intervention, as well as 10 individuals (18%) who also returned to work but with less working time than before the intervention. Six patients (11%) did not return to work after TKA at one year following the surgery [29]. A study by Scott analyzed 289 patients after TKA. A multivariate analysis showed that heavy or moderate physical work and age were the main factors contributing to a return to employment of any kind or a return to the same job. In the group of patients working before TKA, all those below 50 years of age and 60% of those aged 50-54 years returned to work in some capacity [30]. According to the Nordic Arthroplasty Register, the number of TKA interventions doubled over a 15-year period, whereas the number of THA procedures increased by 2.5 times. Due to the strong results of this research, it was possible to expand the indications to younger patients of working age [31]. Patients also continued to work longer as the retirement age increased.

Point 5: Methodology: Some important points of this session could be changed. Study design could be better presented and marked sentences further analyzed. Exclusion and inclusion criteria could be more specific regarding some important topics like measurements including BMI and height of the patients. The way self-assessment of health was evaluated should be defined (whether an already validated scale has been used). When, it is feasible, some additional subgroups could be presented (related with sex, due to different age threshold applied).

Response 5: Thank you for the helpful suggestions. In accordance with the Reviewer’s recommendation, we have enhanced the methodology and design. The aim of the study has been rewritten to make it more specific and accurate. We have revised exclusion and inclusion criteria to make them more detailed. We have added detailed description of the self-assessment of health, with the related reference. We agree that patient’s BMI or height may considerably impact the recovery of functional capacities after TKA. We have addressed this issue in the Study limitations, and we will definitely take these factors into account in our further research. We also agree that distinction of groups related to age and sex is very important, but our sample was narrowed down to individuals with primary and unilateral TKA due to degenerative disease, as a result of which only 48 individuals were taken into account. Division into subgroups would make the process of drawing conclusions more difficult. In future studies we will definitely divide the study group into additional subgroups related with sex, if the same different age threshold is applied. We have addressed this in the Study limitations section.

BEFORE

lines 62-64: Introduction

To our knowledge, no studies so far have investigated the level of participation and RTW after TKA in Poland. Hence, this study aimed to identify predictors of RTW after TKA in patients of working age (≤ 65 years) in Poland.

AFTER

lines 94-99: Introduction

However, there are no reliable data on Polish patients. For this reason, the present retrospective study was designed to determine the RTW rates after TKA, provide reference data, and identify the factors impacting patients’ decisions to return to or resign from work, as well as their functional capacities following TKA. It was hypothesized that RTW rates in patients after TKA in Poland would be similar to those in developed countries.

BEFORE

lines 62-64: Materials and Methods

2.1. Participants

The study was conducted in a group of patients treated at the Department of Orthopaedics and Traumatology of the Regional Clinical Hospital No. 2 in Rzeszów, Poland between 2013 and 2018. All surgical procedures were performed by the same team of five orthopaedists, using the same surgical technique: tibia first and mechanical alignment (TF&MA).

A retrospective analysis of the medical records was performed, taking into account patients meeting eligibility criteria, i.e., individuals with unilateral TKA, women and men aged  ≤ 54 and ≤ 59 years, respectively, on the day of the surgery (in Poland the retirement age for women and men is 60 and 65 years, respectively), individuals over one year after surgery. Exclusion criteria: revision knee arthroplasty and simultaneous bilateral TKA, other musculoskeletal diseases (e.g. spinal stenosis), a history of other surgeries or internal diseases making it impossible to return to work, oncological diseases, musculoskeletal infections and lack of consent to participate.

Cruciate retaining prostheses of two designs were applied, i.e. Triathlon (Stryker Or-thopaedics, Mahwah, New Jersey), and PFC Sigma (Depuy Synthes, Raynham, Massa-chusetts). The patella was not routinely resurfaced.

The procedure applied primary condylar endoprosthesis Columbus (Aesculap BBraun, Tuttlingen, Niemcy) and Vanguard (Biomet, Warsaw, USA) with similar kinematic characteristics. Patellar endoprosthesis was not applied.

In the total group of 431 patients after TKA, inclusion criteria were met by 48 patients, 23 females and 25 males. On average the participants were aged 59 (±4.09) years.

All the patients gave their consent to participate in the study. The study was ap-proved by the Bioethics Commission of the Centre for Post-graduate Medical Training in Warsaw (No. 71/PB/2012).

2.2. Measures

 All the patients selected based on a review of medical records were approached by phone by a doctor, member of the hospital staff and the research team. A phone interview, based on a questionnaire, was conducted with all participants by a member of the team. The 14-item questionnaire was designed by the team of researchers (orthopaedists and experienced physiotherapists). The first part of the questionnaire was related to the demographics and body size measures. The subsequent items asked about the date of surgery, post-operative rehabilitation and post-operative complications. The final part focused on the participants’ occupational status before and after TKA, factors that influenced the decision to return to work and self-assessment of health.

Functional limitations in everyday life due to a knee dysfunction following TKA were assessed using the Knee Outcome Survey - Activities of Daily Living (KOS-ADL) [12]. This is a reliable and accurate instrument that responds to changes in the functional performance of patients with various knee conditions who receive physiotherapy or orthopaedic interventions  [13,14]. The study applied a Polish version of KOS-ADL [15]. Responses are given on a scale of 0-5. The maximum final score is 70 points. To calculate the Activity in Daily Living (ADL) index, the final score is divided by 70 and multiplied by 100. KOS-ADL scores are presented for values normalised to the range 0-100.

2.3. Statistical methods

Calculations were performed using the IBM SPSS programme. For nominal and ordinal variables, results were presented as relative and absolute numbers (% and N). Results for quantitative variables were presented using descriptive statistics (arithmetic mean, median, standard deviation). Significance of the differences in the quantitative variables between groups was assessed using Mann-Whitney test. Assessment of the differences in the nominal variables between the groups was performed using chi-squared test. The Spearman correlation coefficient was used to assess the relationship; the limit of statistical significance in the tests performed was defined as p<0.05.

AFTER

lines 100-177: Materials and Methods

2.1. Participants

This retrospective, single-center study was conducted on a group of patients treated at the Department of Orthopedics and Traumatology of the Regional Clinical Hospital No. 2 in Rzeszów, Poland, between 2013 and 2018. All surgical procedures were performed by the same team of five orthopedists using the same surgical technique: tibia first and mechanical alignment (TF&MA).

A retrospective analysis of the medical records was performed, taking into account patients meeting eligibility criteria, i.e., individuals with unilateral TKA, women aged ≤54 years, and men aged ≤59 years on the day of the surgery. These age criteria for men and women were applied because of the legal retirement age in Poland, which is 60 for women and 65 for men. It was assumed that a period of five years before reaching retirement age and a period of at least one year following TKA would be sufficient for the patients to return to work after TKA. Exclusion criteria included revision knee arthroplasty and simultaneous bilateral TKA, other musculoskeletal diseases (e.g., spinal stenosis), a history of other surgeries or internal diseases making it impossible to return to work, oncological diseases, musculoskeletal infections, and lack of consent to participate.

Because of their similar construction design, kinematics, instrumentation, and corresponding surgical techniques, prostheses from two companies were selected: Columbus™ (Aesculap B Braun, Tuttlingen, Germany) (n=30) and Vanguard (Biomet, Warsaw, USA) (n=18). Both designs are primary, condylar prostheses with over 15 years of orthopedic application and include unconstrained posterior cruciate retaining (CR). In addition, intact PCL provides stability under flexion and allows for femoral rollback.

Polyethylene–metal bearings were applied in all cases, and the same approach was used, i.e., the medial parapatellar approach with classic instrumentation, extracortical fixation for tibia first, and intramedullary femoral instrumentation. The same surgical technique for insertion and theory of component fixation were also applied. All procedures were preceded by preoperative planning in the mediCAD software (mediCAD Hectec GmbH, Landshut, Germany). No patelloplasty was performed.

In the total group of 431 patients after TKA, the inclusion criteria were met by 48 patients, including 23 females and 25 males. On average, the participants were aged 59 (±4.09) years.

All patients gave their consent to participate in the study. The study was approved by the Bioethics Commission of the Center for Post-graduate Medical Training in Warsaw (No. 71/PB/2012).

2.2. Measures

 All patients were selected based on a review of medical records. These patients were approached via phone by a doctor, a member of the hospital staff, and the research team. A phone interview based on a questionnaire was conducted with all participants and led by a member of the team. The 14-item questionnaire was designed by the team of researchers (orthopedists and experienced physiotherapists). The first part of the questionnaire was related to the demographics and body size measurements. The subsequent items asked about the date of surgery, post-operative rehabilitation, and post-operative complications. The final part focused on the participants’ occupational status before and after TKA, factors that influenced the decision to return to work and the self-assessment of health.

The patients were asked to assess their health status on a 5-point Likert-type scale by responding to the following question: “Which term most accurately describes your current health compared to others in the same age group?” Possible responses were as follows: very good (=1), good (=2), moderate (=3), not good (=4), or poor (=5). The scoring system was inverted to make the results easier to read (higher scores indicated better self-rated health) [32].

Functional limitations in everyday life due to a knee dysfunction following TKA were assessed using the Knee Outcome Survey—Activities of Daily Living (KOS-ADL) [33]. This survey is a reliable and accurate instrument that responds to changes in the functional performance of patients with various knee conditions who receive physiotherapy or orthopedic interventions [34,35]. The study applied a Polish version of KOS-ADL [36], and responses were given on a scale of 0-5. The maximum final score was 70 points. To calculate the Activity in Daily Living (ADL) index, the final score was divided by 70 and multiplied by 100. KOS-ADL scores were presented for values normalized to a range of 0-100.

2.3. Statistical methods

Prior to the recruitment procedure (a retrospective analysis of medical records), the analysis of power was computed using Statistica, version 13.1 (StatSoft Poland). The general indicator of effect size for all outcome measures and the power of the study were 0.5, the probability was 0.05, and the maximum error was 10%. The estimated target sample size was 45. Ultimately, 48 participants were enrolled in this study.

Statistical analyses were computed using SPSS (IBM, Chicago, IL, USA), version 21.0. Categorical variables were reported as percentage values, and continuous variables were presented as the mean, standard deviation, and median values. Analysis of the normal distribution was carried out using a Shapiro–Wilk test. Comparative analysis of the KOS-ADL scores after surgery in the occupational activity groups was performed using the non-parametric Mann–Whitney test. The Spearman correlation coefficient was used to assess the relationship between functional performance measured with KOS-ADL and self-rated health status. Statistical significance was confirmed at p < 0.05.

BEFORE

lines 300-310: Study limitations

Despite retrospective recruitment of 431 people, only 48 met the criteria, making further analysis difficult. A multicentre study taking into account different social settings is essential to get an adequate picture of the problem. The study also did not assess the working hours of those who chose to return to work, so the findings do not show whether these participants returned to work full-time or part-time. Another limitation is the fact that no information was acquired with respect to the participants’ rehabilitation process. Indeed, rehabilitation may be greatly varied, ranging from the most basic post-operative therapies, to in-patient programs at hospital rehabilitation units or systematic treatment programs at rehabilitation centres, hence the type of rehabilitation received may be related to the decision about returning to work.

AFTER

lines 426-444: Study limitations

Despite the retrospective recruitment of 431 patients, only 48 individuals met the inclusion criteria, making further analysis difficult. A multicenter study taking into account different social settings is essential to obtain an adequate overview of the problem. The study also did not assess the working hours of those who chose to return to work, so the findings did not indicate whether these participants returned to work full-time or part-time. Another limitation is the fact that no information was acquired on the participants’ rehabilitation process. Indeed, rehabilitation can vary greatly, ranging from the most basic post-operative therapies to in-patient programs at hospital rehabilitation units or systematic treatment programs at rehabilitation centers. Thus, the type of rehabilitation received may be related to one’s decision to return to work. Furthermore, the eligibility criteria in the present study did not include the BMI or height of the patients. One’s BMI and height can considerably impact the recovery of functional capacities after TKA. Thus, these factors should be considered in further research. Future studies should also distinguish between additional sex-related subgroups due to the different age thresholds applied. The division of groups by age and sex is very important, but our sample was narrowed to individuals with primary and unilateral TKA due to degenerative disease, leaving only 48 individuals for the final analysis. For this reason, a further division into subgroups would have made the process of drawing conclusions more difficult.

Point 6: Results: This session could be minimized and information already presented in the tables could be removed from the text, which should be focused on the most important, ones. Results could be divided in subgroups, taking account different age threshold. Marked points could be more clearly presented.

Response 6: Thank you for the helpful suggestions. In accordance with the Reviewer’s comment, we have removed information already presented in the tables from the text. We have added the flow diagram in the Results. Detailed results of KOS-ADL (along with the result of the statistical test) have been presented for the participants of the study relative to the type of work, and for groups of those who did and did not return to work. In the revised manuscript, the results of KOS-ADL assessment are presented relative to self-rated health status in the working and non-working groups after TKA. The findings of correlation analysis for KOS-ADL score and self-rated health as well as the differential analysis of self-rated health in the working and non-working group after TKA have also been added in the Results section. We also agree that distinction of groups related to age and sex is very important, but our sample was narrowed down to individuals with primary and unilateral TKA due to degenerative disease, as a result of which only 48 individuals were taken into account. Division into subgroups would make the process of drawing conclusions more difficult. In future studies we will definitely divide the study group into additional subgroups related with sex, if the same different age threshold is applied. We have addressed this in the Study limitations section.

BEFORE

lines 120-185: Results

3.1. The flow of participants

Based on medical records 431 patients were selected for the study. The age criterion was exceeded by 85.6% of the patients (369 individuals). Following the first contact by phone, 14 individuals were excluded (3 individuals were dead, 3 patients refused to participate, 2 individuals changed their phone number, 6 patients reported they had another TKA).

3.2. Characteristics of the participants

Out of all the participants, 28 patients (58.3%) lived in rural areas and 20 patients (41.7%) lived in urban areas. The participants’ BMI ranged from 20.8 to 41.1. The mean BMI was 31.446 (median value of 31.947). Post-operative complications were reported by 83% of the participants, all patients reported only intermittent pain in the operated lower limb. The participants’ self-reported health was average (41.7%), good (27.1%), poor (16.7%), very good (12.5%) and excellent (2.1%).

Before TKA, 21 individuals (43.75%) in the group performed physical jobs unrelated to agriculture; 15 individuals (31.25%) did not work and were receiving benefits due to osteoarthritis of the knee; nine individuals (18.75%) performed physical work related to agriculture; and office work was performed by three individuals (6.25%). After TKA the group comprising those who did not work and were receiving benefits increased by 23, reaching the number of 38 individuals (79.17%). Other physical jobs were performed by six individuals (12.5%), physical work related to agriculture by two individuals (4.17%) and office work by two individuals (4.17%) – Table 1.

Complications after TKA, including restricted mobility of the knee and pain in the operated lower limb, were reported on the day of the examination by 8.3% of the participants. Assessment of the participants’ current health showed that average health status was reported by 41.7%, good health by 27.1%, and poor health was reported by 16.7% of the participants, whereas 14.6% of the study participants reported very good health.

3.3. Factors impacting a decision to resign from work after the surgery

After the surgery 23 persons did not return to work (47.9% of all those who worked prior to TKA). The reasons for resigning from work, reported by the participants, are shown in Table 2.

Those who did not return to work were asked whether they would go back to work if certain conditions were met. Majority of the participants (76.5%) would not return to work if it was possible for them. The others would take up a job if they had the opportunity to retrain (11.8%) and if the employer removed obstacles at the workplace  (11.8%) - Table 3.

3.4. Assessment of the participants’ functional performance

The mean level of functional performance after TKA, assessed using KOS-ADL was 75.89. The mean KOS-ADL score in the case of the individuals who were receiving welfare benefits prior to TKA and those who did not return to work after TKA and started receiving the benefits was 76.84. The mean KOS-ADL score in the case of the individuals who worked prior to TKA and those who returned to work after TKA was 75.57. The difference was not statistically significant (p=0.055).

The highest score, amounting to 91.7, was achieved by the participants who returned to physical work after the surgery. These were followed by the participants who retired: mean of 74.25 as well as those performing physical work in agriculture: mean and median value of 74%. The poorest score were acquired by the participants performing white-collar jobs, with the mean score of 61.0 (Table 4).

AFTER

lines 178-268: Results

3.1. The flow of participants

Based on medical records, 431 patients were selected for this study. The age criterion was exceeded by 85.6% of the patients (369 individuals). Following the first discussion by phone, fourteen individuals were excluded (three individuals had died, three patients refused to participate, two individuals changed their phone number, and six patients reported that they had another TKA). The detailed flow of participants in the study is shown in Figure 1.

3.2. Characteristics of the participants

Out of all the participants, 28 patients (58.3%) lived in rural areas, and 20 patients (41.7%) lived in urban areas. The participants’ BMI ranged from 20.8 to 41.1. The mean BMI was 31.446 (median value of 31.947). Post-operative complications were reported by 83% of the participants, and all patients reported only intermittent pain in the lower limb that was operated upon. In descending order of frequency, the participants’ self-reported health was average (41.7%), good (27.1%), poor (16.7%), very good (12.5%), and excellent (2.1%). The occupational activities of the participants before and after TKA are shown in Table 1.

Complications after TKA, including restricted mobility of the knee and pain in the lower limb that was operation on, were reported on the day of the examination by 8.3% of the participants. In the assessment of current health, average health status was reported by 41.7%, good health by 27.1%, and poor health by 16.7% of participants; only 14.6% of the study participants reported very good health.

3.3. Factors impacting a decision to resign from work after the surgery

After the surgery, 23 patients did not return to work (47.9% of all those who worked prior to TKA). The reasons for resigning from work, as reported by the participants, are shown in Table 2.

Those who did not return to work were asked whether they would go back to work if certain conditions were met. The majority of the participants would not return to work if they could. The others responded that they would take a job if they could retrain and/or if obstacles in the workplace were removed (see Table 3).

3.4. Assessment of the participants’ functional performance

The mean level of functional performance after TKA assessed using KOS-ADL was 75.89. The mean KOS-ADL score was 76.84 among individuals who were receiving welfare benefits prior to TKA, did not return to work after TKA, and started receiving benefits only after the surgery. The mean KOS-ADL score among individuals who worked prior to TKA and returned to work after TKA was 75.57. This difference was not statistically significant (p=0.055). The lowest level of functional performance was found in the group of individuals doing office work (61.0), while the highest was observed among those doing physical jobs (91.7). The individuals doing physical work in agriculture achieved an average score of 74.0 in KOS-ADL. No statistically significant differences were observed between the groups (p=0.055) (Table 4).

The participants were divided into two groups: those who worked after TKA (regardless of the type of job) and those who did not return to work. The mean KOS-ADL scores in the former and latter groups were 83.67 and 76.67, respectively. This difference was not statistically significant (p=0.55).

The findings show a moderate correlation (r=4.24; p=0.003) between functional performance in KOS-ADL and self-rated health status. No statistically significant differences in self-rated health status were found between those who did and did not return to work after TKA (p=0.34).

BEFORE

lines 300-310: Study limitations

Despite retrospective recruitment of 431 people, only 48 met the criteria, making further analysis difficult. A multicentre study taking into account different social settings is essential to get an adequate picture of the problem. The study also did not assess the working hours of those who chose to return to work, so the findings do not show whether these participants returned to work full-time or part-time. Another limitation is the fact that no information was acquired with respect to the participants’ rehabilitation process. Indeed, rehabilitation may be greatly varied, ranging from the most basic post-operative therapies, to in-patient programs at hospital rehabilitation units or systematic treatment programs at rehabilitation centres, hence the type of rehabilitation received may be related to the decision about returning to work.

AFTER

lines 426-444: Study limitations

Despite the retrospective recruitment of 431 patients, only 48 individuals met the inclusion criteria, making further analysis difficult. A multicenter study taking into account different social settings is essential to obtain an adequate overview of the problem. The study also did not assess the working hours of those who chose to return to work, so the findings did not indicate whether these participants returned to work full-time or part-time. Another limitation is the fact that no information was acquired on the participants’ rehabilitation process. Indeed, rehabilitation can vary greatly, ranging from the most basic post-operative therapies to in-patient programs at hospital rehabilitation units or systematic treatment programs at rehabilitation centers. Thus, the type of rehabilitation received may be related to one’s decision to return to work. Furthermore, the eligibility criteria in the present study did not include the BMI or height of the patients. One’s BMI and height can considerably impact the recovery of functional capacities after TKA. Thus, these factors should be considered in further research. Future studies should also distinguish between additional sex-related subgroups due to the different age thresholds applied. The division of groups by age and sex is very important, but our sample was narrowed to individuals with primary and unilateral TKA due to degenerative disease, leaving only 48 individuals for the final analysis. For this reason, a further division into subgroups would have made the process of drawing conclusions more difficult.

Point 7: Discussion: An extended presentation of the literature regarding the points under investigation. Some paragraphs of this session could be incorporated in the introduction session and some others could be removed in order to enhance conclusions session.  Paragraphs analyzing known literature could be shortened and these used to explain the results of this study could be presented with tables, summarizing results. An interesting comparison of KOOS-ADL with known literature is applied , but the comparison with the pre-operative values could be useful, if feasible.

Moreover, there are existing studies in the literature on TKA that investigate the correlations between KOOS and other patient-reported outcomes with various parameters. Exploring these papers could be beneficial for your discussion section.

Response 7: Thank you for the helpful suggestions. We have revised the discussion, addressing the important issues related to the impact of rehabilitation and the relationship between RTW and functional performance. Some paragraphs previously in this section have been incorporated in the Introduction and some others have been removed in order to enhance the conclusions section. 

BEFORE

lines 186-310: Discussion

The study asked three questions: how many people return to work after TKA one year after surgery, what factors contribute to the decision to resign from work and what is the level of functional performance in the study participants.

The initial assessment involved a large group of 431 patients after unilateral TKA due to knee osteoarthritis. It was shown that as many as 85.6% of the individuals in the group did not meet the age criterion. Vast majority of the patients had reached the retirement age by the day of the surgery. The participants’ age on the day of TKA was similar to the data in the  National Joint Registry (NJR), where those aged 50-59 years in 2020 accounted for 15% of the total 53,782 individuals after TKA [6]. This is an important issue in the context of increasing life expectancy and the need to continue working in order to maintain quality of life both in material and social terms. However, the present study does not answer an important question, namely how many of those excluded from the study had discontinued gainful employment and started receiving welfare benefits due to osteoarthritis before reaching the retirement age. This is important in the context of the selected timing of the intervention and the potential return to work. 

In 2014 an exhaustive systematic review was published by Tilbury et al., who identified 19 articles focusing on RTW after TKA and/or Total Hip Arthroplasty (THA), with 14 articles related to THA and 4 focusing on TKA only. The percent of patients returning to work was in the range between  25% - 95% in the period from 1 to 12 months after THA, and in the range between 71% and 83% in the period from 3 to 6 months after TKA [8].

Lichtenberg reported in 2016 that in a post-TKA group of 56 individuals aged below 65 years (mean 56 years), there were 40 individuals (71%) who worked prior to the surgery and returned to full-time work one year after the intervention as well as 10 individuals (18%) who also returned to work but with less working time than before the intervention. Six patients (11%) after TKA did not return to work at one year following the surgery [10].

According to the Nordic Arthroplasty Register, the number of TKA interventions doubled over a 15-year period, whereas the number of THA procedures increased 2.5 times. Owing to the good results it was possible to expand the indications to younger patients of working age [16] . Furthermore, patients continue to work longer as the retirement age is increased. There are few studies focusing on RTW after THA and TKA, and none in France. The average time to return to work was found to be between 8 and 12 weeks following TKA. [17,18], whereas the RTW rates are in the range between 40% and 98% in the case of TKA [19]. Mongin  investigated RTW rates at one year following TKA or THA in a group of 299 individuals below 65 years of age. In that group, 144 individuals worked up until the day of the surgery. After TKA, 38 patients returned to work (52.7%) on average after 141±100.69 days, and 34 did not resume work [20]. In the present study, 31.25% of the individuals included did not work before the surgery. Those not working were receiving welfare benefits (pension) due to the degenerative joint disease of the knee. The group of those who did not work after TKA increased significantly, to a level of 79.17%. All of the non-working participants were receiving a pension, due to the fact that they were unable to perform their previous job and presented low level of fitness. Only 20.83 % of participants therefore returned to work. This is a very low percentage compared to the results of many studies where RTW rates after TKA are between 40% and 98%.

Styron showed that the factors linked to faster RTW after TKA included female gen-der, self-employment, better post-operative physical and mental health outcomes, higher functional comorbidity index and a workplace accessible for disabled people. Slower RTW after TKA was linked to the preoperative level of pain, more physically demanding work, and employee’s compensation [21]. Bardgett identified factors that affected the RTW pro-cess after TKA. According to patients, these factors included delays in surgical intervention, limited and often inconsistent RTW-related advice from healthcare professionals, and finally a lack of rehabilitation to optimise the patient's recovery and facilitate RTW [22].

The present study provides quantitative information on the RTW rate after TKA. To date, only very few studies in Poland (Pop and Truszyńska) have reported data related to RTW after hip replacement, whereas no studies have focused on RTW after TKA. A retrospective study by Foote involving 109 individuals under 60 years of age showed that 82% of patients who had worked before surgery and who had patellofemoral replacement or unicompartmental replacement returned to work within 12 weeks. Following TKA, 54% of the patients returned to work within 20 weeks after the surgery. In the present study, RTW rate after TKA was 30.3%. Before TKA, 31.25% of the patients were receiving pension due to osteoarthritis of the knee. After the intervention, 79.16% were receiving pension due to the disease and post-operative condition. The low RTW rate in the study group may be linked to a number of reasons. Individuals performing physical jobs, including agricultural workers, accounted for  62.5% of the group. Those who did not return to work after TKA pointed out that the main reason for giving up work was the inability to perform the job they had before the surgery and the lack of opportunity to retrain or find another job (58.8%). Poor self-assessed health after TKA was the second most important reason for giving up work, reported by 23.5% of the participants. Only two participants (11.8%) re-ported that the opportunity to receive welfare benefits was the reason for giving up work, whereas 5.9% of the participants reported a lack of acceptance from the employer as the reason. It is concerning that in response to the question whether they were informed about the possibility of retraining, the participants who provided answer reported that they had not been given any such information before or after TKA. However, even if certain conditions were met, for instance if there were opportunities for retraining or if the situation was accepted by the employer, majority of the participants stated that they would not take up a job after TKA. Poor health after TKA was reported as a reason to resign from work by 23.5% of the participants. It is interesting to analyse the results of the participants’ functional assessment and to compare the KOS-ADL score of those who returned to work with those who did not. The difference was not statistically significant, so the decision about RTW was impacted not only by the fitness level but by other factors, probably social. There are conflicting opinions about the relationship between functional capacity and RTW after TKA. Leichtenberg assessed preoperative functional capacity with the KOS-ADL scale and found that there was a relationship between KOS-ADL score and RTW after TKA [10] In contrast, Kuijer did not observe a significant relationship between KOS-ADL score and RTW [26].

Currently, there is no consistent process for optimising RTW of working-age patients after TKA. The effects of late surgical intervention, as well as a lack of or limited RTW-related counselling contribute to potential delays in return to work [23,24,25]. Therefore, it is necessary to refocus healthcare provision for this group of patients, particularly with regard to RTW-oriented rehabilitation informing about the possibility to per-form work and containing tailored interventions to optimise patient outcomes [22].

Patients taking part in this study indicated a need for counselling related to their engagement in work that would match their specific employment requirements, which would potentially be a factor initiating their return to work, but, on the other hand, they also admitted that they did not want to return to work after TKA. The study group mainly included physical workers and agricultural workers, i.e., individuals with limited options for retraining, who are also unable to continue to perform physically demanding jobs due to the surgery.

It is also necessary to emphasize the importance of the welfare benefits that most of the patients received after TKA since this factor seems to encourage patients to withdraw from the labour market.

Tilbury points out, in reference to the studies carried out to date in various countries, that it cannot be ruled out that the timing of RTW may be affected by the health care system or welfare security system. As an example he reports that the sick leave from work in the Netherlands is paid in full for the first two years [27]. Lombardi points out that in the USA and other countries, affordable health insurance is an advantage and refers to a study in which 77% of the individuals aged 18-69 years post-TKA obtained benefits from health insurance provided by their employer [19]. Perhaps the low RTW rate identified in this study may be linked to the Polish healthcare and social security system.

Study limitations

Despite retrospective recruitment of 431 people, only 48 met the criteria, making further analysis difficult. A multicentre study taking into account different social settings is essential to get an adequate picture of the problem. The study also did not assess the working hours of those who chose to return to work, so the findings do not show whether these participants returned to work full-time or part-time. Another limitation is the fact that no information was acquired with respect to the participants’ rehabilitation process. Indeed, rehabilitation may be greatly varied, ranging from the most basic post-operative therapies, to in-patient programs at hospital rehabilitation units or systematic treatment programs at rehabilitation centres, hence the type of rehabilitation received may be related to the decision about returning to work.

AFTER

lines 271-444: Discussion

The present study asked three questions: How many people return to work after TKA one year after surgery, what factors contribute to the decision to resign from work, and what is the level of functional performance among the study participants?

The initial assessment involved a large group of 431 patients who received unilateral TKA due to knee osteoarthritis. It was shown that as many as 85.6% of the individuals in the group did not meet the age criterion. The vast majority of patients had reached the retirement age by the day of the surgery. The participants’ ages on the day of TKA agreed with the data in the National Joint Registry (NJR), where those aged 50-59 years in 2020 accounted for 15% of 53,782 total individuals after TKA [25]. This is an important issue due to increasing life expectancy and the need for individuals to continue working in order to maintain quality of life in both material and social terms. However, the present study did not answer an important question, namely, how many of those excluded from the study had discontinued their gainful employment and started receiving welfare benefits due to osteoarthritis before reaching retirement age. This is important in the context of the selected timing of the intervention and the potential return to work.

There are few studies focusing on RTW after TKA. The average time to return to work was found to be between 8 and 12 weeks following TKA [37,22], whereas the RTW rates were between 40% and 98% in the case of TKA [38]. Mongin investigated the RTW rates at one year following TKA or THA in a group of 299 individuals below 65 years of age. In that group, 144 individuals worked until the day of the surgery. After TKA, an average of 38 patients returned to work (52.7%) after 141±100.69 days, and 34 did not resume work [39]. In the present study, 31.25% of the individuals included did not work before the surgery. Those not working were receiving welfare benefits (a pension) due to a degenerative joint disease of the knee. The group of those who did not work after TKA increased significantly to a level of 79.17%. All non-working participants were receiving a pension because they were unable to perform their previous job and presented a low level of fitness. Only 20.83% of participants, therefore, returned to work. This is a very low percentage compared to the results of many studies in which RTW rates after TKA were between 40% and 98%.

The present study provides quantitative information on the RTW rates after TKA. To date, very few studies in Poland have reported data related to RTW after hip replacement [40], and no studies have focused on RTW after TKA. A retrospective study by Foote involving 109 individuals under 60 years of age showed that 82% of patients who had worked before surgery and received patellofemoral replacement or unicompartmental replacement returned to work within 12 weeks. Following TKA, 54% of the patients returned to work within 20 weeks after surgery [35]. In the present study, the RTW rate after TKA was 30.3%. Before TKA, 31.25% of the patients were receiving a pension due to osteoarthritis of the knee. After the intervention, 79.16% were receiving a pension due to the disease and their post-operative condition. The low RTW rate in the study group may be due to a number of reasons. Individuals performing physical jobs, including agricultural workers, accounted for 62.5% of the group. Those who did not return to work after TKA noted that their main reason for resigning from work was the inability to perform the job they had before the surgery and the lack of opportunities to retrain or find another job (58.8%). Poor self-assessed health after TKA was the second most important reason for giving up work and was reported by 23.5% of the participants. Only two participants (11.8%) reported that the opportunity to receive welfare benefits was their reason for giving up work, whereas 5.9% of the participants reported a lack of acceptance from their employer as the reason. When asked whether they were informed about the possibility of retraining, the participants who provided answers concerningly reported that they had not been given any such information before or after TKA. Nevertheless, even if certain conditions were met, such as opportunities for retraining or greater employer acceptance, the majority of the participants stated that they would not take a job after TKA. In addition, poor health after TKA was reported by 23.5% of the participants as a reason to resign from work. Achieving the highest possible level of functional capacity after TKA is the main goal of therapy. We hypothesized that individuals who returned to work after TKA would present a higher level of functional performance than those who did not return to work after TKA. However, our findings presented a different picture. Those who did not work reported fewer limitations due to pain, swelling, and joint stiffness, as well as fewer limitations related to gait and basic gait function, compared to those who returned to work. Overall, the difference in functional performance between the two groups was not statistically significant. Therefore, other factors may have impacted the decision to return to or resign from work among the study group. Since no assessment with KOS-ADL was performed before TKA, it was impossible to assess the effects of changes in functional performance on RTW after surgery. However, there are conflicting opinions about the relationship between functional performance and RTW after TKA. Leichtenberg assessed pre-operative functional performance using KOS-ADL and reported a positive relationship between the KOS-ADL score and RTW after TKA [29]. Conversely, Kuijer did not find a significant relationship between the KOS-ADL score and RTW [41]. In the group of individuals who returned to work, those performing office jobs presented a lower rating of functional performance than that reported by patients who performed physical work in agriculture or other sectors. Consequently, there are no grounds for claiming that physical activity, including rehabilitation, impacted the perception of functional performance in the study group. However, there remains a need for further research to reliably assess the impact of rehabilitation on RTW after TKA.

Currently, there is no consistent process for optimizing RTW among working-age patients after TKA. The effects of late surgical intervention and absent or limited RTW-related counselling contribute to potential delays in return to work [42-44]. Therefore, it is necessary to refocus healthcare provisions for this group of patients, particularly with regard to RTW-oriented rehabilitation. Such provisions could inform patients about the possibility to perform work and contain tailored interventions to optimize patient outcomes [23]. A previous study showed that the process of recovery after TKA might benefit from high-volume exercise applied before surgery. However, assessments of exercise-based interventions applied after surgery did not show the advantage of one exercise program over another [45]. Therefore, there is no clear evidence that rehabilitation after TKA based on a specific type of activity would generally enhance recovery in patients [46,47]. It appears that earlier and more intense physical rehabilitation programs involving exercise (e.g., strength training) produce better effects than less intense programs (e.g., ADL training without strength training) [48]. The most recent reports concerning the impacts of various rehabilitation strategies on RTW outcomes suggest that specific rehabilitation interventions should primarily consider patients' expectations for their post-operative recovery, their willingness to engage in exercise and physical activity, and their pre-operative functional performance [45]. Therefore, future strategies should emphasize the importance of pre-operative exercises (pre-habilitation) in patients with reduced pre-operative knee extension strength, impaired functional performance, and a tendency to fall [45].

Patients taking part in this study indicated a need for counselling related to finding work that would match their specific employment requirements. Participants noted that such information could potentially prompt a return to work but also admitted that they did not want to return to work after TKA. The study group mainly included physical workers and agricultural workers, i.e., individuals with limited options for retraining who were also unable to continue to perform physically demanding jobs due to the surgery.

It is also necessary to emphasize the importance of the welfare benefits that most of the patients received after TKA since this factor seemed to encourage patients to withdraw from the labor market.

In reference to studies carried out to date in various countries, Tilbury noted that the timing of RTW may be affected by the health care and welfare security systems. As an example, Tilbury reported that sick leave from work in the Netherlands is paid in full for the first two years [49]. Lombardi noted that in the USA and other countries, affordable health insurance is an advantage, highlighting a study in which 77% of individuals aged 18-69 years post-TKA obtained benefits from the health insurance provided by their employer [38]. The low RTW rate identified in this study may be linked to the Polish healthcare and social security system. In Poland, welfare benefits are commonly awarded to individuals incapable of working due to a degenerative disease or surgery. Given the relationship between minimum wage and the amount of these benefits, welfare may be an acceptable solution for many people, especially physical workers with low occupational qualifications. Lombardi suggested that one of the main reasons for not returning to work after TKA is retirement and the use of disability benefits prior to surgery [38]. The present findings support this opinion. As many as 31% of the working-age participants in this study received benefits even before the surgery. After TKA, this group increased significantly.

The presented findings highlighted factors affecting RTW after TKA that are consistent with those reported in other studies. To answer the main question, the group of participants was narrowed to those who were of working age according to Polish regulations and had received surgery within the last year. Unfortunately, the present findings differ significantly from those most frequently reported in other studies. This difference may be due to the small number of participants, but it should be noted that as many as 85% of sampled individuals did not meet the age criterion. The reasons for the low rate of RTW in the study group included a lack of information about the possibility to return to work after surgery, various limitations, and types of work. In Poland, the system of occupational rehabilitation is insufficiently developed. Information on opportunities for retraining or finding another job is not made available to patients with developing osteoarthritis or during the pre-operative period. The most common route for the treatment and rehabilitation process is surgery; rehabilitation; satisfactory recovery of functional capacities; and, as demonstrated here, the acquisition of an incapacity pension. This process was confirmed by the findings of our study, which showed that functional performance after TKA was similar between those who returned to work and those that did not.

Considering the growing number of knee replacement surgeries performed on working-age patients, as well as the increasing life expectancy and corresponding need to increase the duration of one’s working life, this group of patients should receive coordinated medical and rehabilitation care. Considering the importance of returning to work among those who have not yet reached retirement age, further large-scale studies should be conducted considering environmental factors and the quality of life among individuals after TKA. It is also necessary to conduct research evaluating the effectiveness of interventions applied in occupational rehabilitation and functional training after TKA. Over the past few years, high hopes have been placed on the modification of surgical technology using computer navigation and robotics. Indeed, new techniques may reduce the number of patients reporting post-operative problems that prevent them from returning to work. Successful RTW is increasingly recognized as an important outcome for this group, with social and economic consequences for patients, employers, and society.

Study limitations

Despite the retrospective recruitment of 431 patients, only 48 individuals met the inclusion criteria, making further analysis difficult. A multicenter study taking into account different social settings is essential to obtain an adequate overview of the problem. The study also did not assess the working hours of those who chose to return to work, so the findings did not indicate whether these participants returned to work full-time or part-time. Another limitation is the fact that no information was acquired on the participants’ rehabilitation process. Indeed, rehabilitation can vary greatly, ranging from the most basic post-operative therapies to in-patient programs at hospital rehabilitation units or systematic treatment programs at rehabilitation centers. Thus, the type of rehabilitation received may be related to one’s decision to return to work. Furthermore, the eligibility criteria in the present study did not include the BMI or height of the patients. One’s BMI and height can considerably impact the recovery of functional capacities after TKA. Thus, these factors should be considered in further research. Future studies should also distinguish between additional sex-related subgroups due to the different age thresholds applied. The division of groups by age and sex is very important, but our sample was narrowed to individuals with primary and unilateral TKA due to degenerative disease, leaving only 48 individuals for the final analysis. For this reason, a further division into subgroups would have made the process of drawing conclusions more difficult.

Point 8: Conclusions: Conclusions should be focused on what the results of this study can offer to the clinicians regarding the results of TKAs and especially return to work. This session could be further enhanced with comments from the discussion, too.

Response 8: Thank you for the helpful suggestions. In accordance with the Reviewer’s recommendation, we have improved the Conclusion section. In line with the Reviewer’s comment, we have revised the conclusions adding information about the implications of these findings for the clinicians, with regard to the results of TKA and patients’ return to work.

BEFORE

lines 311-326: Conclusions

In view of the growing number of knee replacement surgeries performed in working-age patients, as well as the increasing life expectancy and the resulting need to in-crease the duration of working life it appears that this group of patients needs to receive coordinated medical and rehabilitation care. Bearing in mind how important it is to return to work for people who have not reached the retirement age, further large-scale studies should be conducted taking into account environmental factors and the quality of life in individuals after TKA. It is also necessary to conduct research evaluating the effectiveness of interventions applied in occupational rehabilitation and functional training after TKA. Over the past few years, high hopes have been placed on the modification of surgical technology using computer navigation, or robotics. It is likely that the new techniques may reduce the number of patients reporting post-operative problems which prevent them from returning to work. A successful RTW is more and more commonly recognised as an important outcome for this group, with social and economic consequences for patients, employers and society.

AFTER

lines 445-454: Conclusions

This study was conducted due to the increasing number of patients receiving TKA for knee osteoarthritis and the associated medical and social problems. The findings show that the rate of RTW after TKA in Poland is significantly lower than that in other countries. The reasons for this situation, as shown in the study, may be related to the lack of an occupational rehabilitation system, resulting in a paucity of information about the possibility to return to work and opportunities for retraining. The results of this study suggest a significant deficit in occupational rehabilitation programs designed for individuals who have regained a significant degree of functional capacity after TKA due to knee osteoarthritis.

Point 9: Comments on the Quality of English Language: moderate english editing is needed

Response 9: Thank you for the helpful suggestions. In accordance with the suggestion, the manuscript has been proofread by MDPI English Language Editing Services and we have received the certificate of language editing use.

Reviewer 3 Report

Comments and Suggestions for Authors

Abstract: please add number (%) for each number in results as it's totally difficult to follow. 

No need to write 431 patients as authors evaluated only 48 patients , it's misleading.

Give a clear conclusion other than recommendation in your abstract please

you need to add KOS-ADL results in your abstract as you mentioned it.

introduction: very long , try to shorten it a little bit if possible

Materials methods / results : Clear

Discussion: better in the new version

conclusion Clear. 

in Summery, new version is significantly better than old version, only abstract should be refined. 

Author Response

Dear Reviewer,        

We thank you for reviewing our article titled, “Return to work after primary total knee arthroplasty: the first Polish retrospective study”. We have made every effort to improve our manuscript, as guided by the reviewer’s helpful suggestions.

We thank the reviewer of all the comments. Answers are summarised below. All changes are highlighted as red text in the manuscript.

We hope you will be pleased with the changes, and support the publication of our revised manuscript.

With kind regards,

The authors of the article

Point 1: Abstract: please add number (%) for each number in results as it's totally difficult to follow.

Response 1: Thank you for the helpful suggestions. In accordance with the Reviewer’s comment we have revised the results part of Abstract.

BEFORE

line 17-20: Abstract

Results: Before TKA, 31.25% of those qualified for the study did not work and were receiving welfare benefits. After the surgery, 23 individuals did not return to work (47.9% of all those working prior to TKA). The group of those who did not work after TKA increased significantly, reaching 79.17%.

AFTER

line 17-20: Abstract

Results: Before TKA, 15 individuals (31.25%) qualified for the study did not work and were receiving welfare benefits. After the surgery, 23 individuals (47.9% of those working prior to TKA) did not return to work. The number of those who did not work after TKA increased to 38 (79.17%), which was a significant change. The mean level of functional performance after TKA assessed using KOS-ADL was 75.89.

Point 2: No need to write 431 patients as authors evaluated only 48 patients , it's misleading.

Response 2: Thank you for the helpful suggestions. In accordance with the Reviewer’s comment we have deleted the sentence.

BEFORE

line 14-17: Abstract

Methods: This retrospective study analyzed 431 patients. Ultimately, 48 individuals were enrolled. For this study, an interview on RTW was carried out. This interview also explored the factors impacting a patient’s decision to return to or resign from work. Functional performance was assessed using the Knee Outcome Survey Activities of Daily Living (KOS-ADL).

AFTER

line 14-16: Abstract

Methods: This retrospective involved 48 patients. An interview related to RTW was carried out to identify the factors impacting a patient’s decision to return to or resign from work. Functional performance was assessed using the Knee Outcome Survey Activities of Daily Living (KOS-ADL).

Point 3: Give a clear conclusion other than recommendation in your abstract please

Response 3: Thank you for the helpful suggestions. In accordance with the Reviewer’s comment we have revised the conclusions in the Abstract.

BEFORE

line 20-23: Abstract

Conclusions: Considering the importance of returning to work for those who have not reached retirement age, further large-scale studies should be conducted while taking into account environmental factors and the quality of life among individuals after TKA

AFTER

line 20-24: Abstract

Conclusions: The findings show that the rate of RTW after TKA in Poland is significantly lower than that in other countries. The reasons for this situation, as shown in the study, may be related to the lack of an occupational rehabilitation system, resulting in a paucity of information about the possibility to return to work and about opportunities for retraining.

Point 4: you need to add KOS-ADL results in your abstract as you mentioned it.

Response 4: Thank you for the helpful suggestions. In accordance with the Reviewer’s comment we have added KOS-ADL results.

BEFORE

line 17-20: Abstract

Results: Before TKA, 31.25% of those qualified for the study did not work and were receiving welfare benefits. After the surgery, 23 individuals did not return to work (47.9% of all those working prior to TKA). The group of those who did not work after TKA increased significantly, reaching 79.17%.

AFTER

line 17-20: Abstract

Results: Before TKA, 15 individuals (31.25%) qualified for the study did not work and were receiving welfare benefits. After the surgery, 23 individuals (47.9% of those working prior to TKA) did not return to work. The number of those who did not work after TKA increased to 38 (79.17%), which was a significant change. The mean level of functional performance after TKA assessed using KOS-ADL was 75.89.

Point 5: introduction: very long , try to shorten it a little bit if possible

Response 5: Thank you for the helpful suggestions. We have revised the Introduction to make it shorter.

BEFORE

line 28-99: Introduction

Knee osteoarthritis is a concern for increasingly more people. This affliction leads to significant impairment of functional capacities, which adversely affects activity and participation, gradually leading to a decrease in quality of life [1,2]. Total Knee Arthroplasty (TKA) is the most commonly applied intervention. This treatment is highly effective since it relieves pain and restores the function of the joint, consequently enabling improvements in the patients’ quality of life [3,4]. Although numerous studies have shown that TKA is one of the most successful and effective procedures in orthopedics [5-11], the outcomes can vary based on several factors, such as patient health status, including mental health, as well as expectations, comorbidities, age, preoperative pain ratings, preoperative levels of physical activity, and rehabilitation adherence [12-19]. Additionally, it was suggested that the design of post-TKA rehabilitation should take into account specific patient risk factors, including age, health status, social support status, and knee-related physical functioning before the surgery [20,21]. Furthermore, Styron showed that the factors linked to a faster return to work (RTW) after TKA included female gender, self-employment, better post-operative physical and mental health outcomes, a higher functional comorbidity index, and an accessible workplace accessible. A slower RTW after TKA was linked to the preoperative level of pain, more physically demanding work, and employee compensation [22]. Bardgett identified factors that affected the RTW process after TKA. According to patients, these factors included delays in surgical intervention, limited and often inconsistent RTW-related advice from healthcare professionals, and a lack of rehabilitation to optimize the patient's recovery and facilitate RTW [23].

According to the data from the National Health Fund, in Poland, as many as 37,821 knee replacement procedures were performed in 2020, including 30,615 TKA procedures. Primary knee arthroplasty was performed in 94.3% of the cases, and 95% of the patients were operated on due to primary gonarthrosis. Women accounted for 72% of all patients who received this treatment, and 80% of the patients were aged 60-79 years. On average, the female and male patients were aged 69 and 67 years, respectively. One in two patients who received a knee replacement in 2019 were 69 years of age or younger [24]. According to data published by The United Kingdom National Joint Registry, in 2016, patients up to 65 years of age accounted for 40% of cases [25]. Notably, the number of patients with diagnosed knee osteoarthritis is consistently growing. Similarly, the number of TKA procedures performed annually is increasing, and individuals up to 65 years of age are expected to be the largest group among those receiving this treatment [26].

Considering that the duration of people’s working lives is increasing, return to work (RTW) after TKA is an issue of great importance. Due to the scarcity of related research, the determinants of RTW after TKA are still unclear. It also remains difficult to compare and analyze the results because the definitions of work status vary widely, and different timeframes are often used to measure this status. In 2014, Tilbury presented the findings of a systematic review of studies investigating RTW after hip or knee replacement [27]. This review showed that the percentage of patients returning to work was between 25% and 95% in the period from 1 to 12 months after THA and between 71% and 83% from 3 to 6 months after TKA [27]. Furthermore, Van Leemput reported that the percentage of individuals working after TKA was between 36% and 89%. Those who worked before and returned to work after TKA accounted for 40% to 98% of the populations studied, whereas the mean duration of the rehabilitation period after TKA ranged from 7.7 to 16.6 weeks [28]. According to Van Leemput, a slower or no RTW after TKA was more likely to be observed in patients performing jobs of a more physical nature or among those who were absent from work before the surgery. However, most patients subjected to TKA returned to work after the procedure [28]. Leichtenberg established that a more advanced age, lower educational level, and preoperative absence from work were associated with a partial or no return to work [29]. More specifically, Leichtenberg reported that in a post-TKA group of 56 individuals aged below 65 years (mean 56 years), there were 40 individuals (71%) who worked prior to the surgery and returned to full-time work one year after the intervention, as well as 10 individuals (18%) who also returned to work but with less working time than before the intervention. Six patients (11%) did not return to work after TKA at one year following the surgery [29]. A study by Scott analyzed 289 patients after TKA. A multivariate analysis showed that heavy or moderate physical work and age were the main factors contributing to a return to employment of any kind or a return to the same job. In the group of patients working before TKA, all those below 50 years of age and 60% of those aged 50-54 years returned to work in some capacity [30]. According to the Nordic Arthroplasty Register, the number of TKA interventions doubled over a 15-year period, whereas the number of THA procedures increased by 2.5 times. Due to the strong results of this research, it was possible to expand the indications to younger patients of working age [31]. Patients also continued to work longer as the retirement age increased.

However, there are no reliable data on Polish patients. For this reason, the present retrospective study was designed to determine the RTW rates after TKA, provide reference data, and identify the factors impacting patients’ decisions to return to or resign from work, as well as their functional capacities following TKA. It was hypothesized that RTW rates in patients after TKA in Poland would be similar to those in developed countries.

AFTER

lines 28-88: Introduction

Knee osteoarthritis is a concern for increasingly more people. This affliction leads to significant impairment of functional capacities, which adversely affects activity and participation, gradually leading to a decrease in the quality of life [1,2]. Total Knee Arthroplasty (TKA) is the most commonly applied intervention. This treatment is highly effective since it relieves pain and restores the function of the joint, consequently enabling improvements in the patients’ quality of life [3,4]. The success of the intervention is often measured by how well a person can resume their normal activities,  and this may be linked to TKA kinematics and biomechanics. Poor kinematics and biomechanics after surgery can lead to issues such as pain, instability, and decreased range of motion, which can hinder a person's capability to perform various activities effectively. In contrast, proper alignment and functioning of the implant can improve the person's mobility and performance, increasing the likelihood of a successful recovery of various functions necessary for social participation [5-11]. Although numerous studies have shown that TKA is one of the most effective procedures in orthopedics [12-18], the outcomes can vary based on several factors, such as patient health status, including mental health, as well as expectations, comorbidities, age, preoperative pain ratings, preoperative levels of physical activity, and rehabilitation adherence [19-26].

Effectiveness of TKA can also be evaluated taking into account patients’ ability to return to work (RTW). In 2014, Tilbury in a systematic review the related literature showed that RTW rate at 3 to 6 months after TKA ranged between 71% and 83% [27]. Another systematic review, by Van Leemput, showed that RTW rate after TKA ranged from 36% to 89%. Those who worked before and returned to work after TKA accounted for 40% to 98% of the populations studied [28]. Given these significant differences in the reported statistics, it seems necessary to identify the factors impacting patients’ decisions about RTW after TKA. The study by Styron showed that a faster RTW after TKA was linked to female gender, self-employment, better post-operative physical and mental health outcomes, a higher functional comorbidity index, and an accessible workplace, whereas a slower RTW after TKA was related to the preoperative level of pain, more physically demanding work, and employee compensation [29]. Furthermore, according to Bardgett the RTW process after TKA was affected by delays in surgical intervention, limited and often inconsistent RTW-related advice from healthcare professionals, and a lack of rehabilitation to optimize the patient's recovery and facilitate RTW [30]. According to Van Leemput, a slower or no RTW after TKA was more likely to be observed in patients performing jobs of a more physical nature or among those who were absent from work before the surgery [28]. Leichtenberg established that a more advanced age, lower educational level, and preoperative absence from work were associated with a partial or no return to work [31]. A multivariate analysis of data collected from 289 patients by Scott et al. showed that heavy or moderate physical work and age were the main factors contributing to a return to employment of any kind or a return to the same job. In the group of patients working before TKA, all those younger than 50 years of age and 60% of those aged 50-54 years returned to work in some capacity [32].

According to the data from the National Health Fund, in Poland, as many as 37,821 knee replacement procedures were performed in 2020, including 30,615 TKA procedures. On average, the female and male patients were aged 69 and 67 years, respectively. Approximately 50% of those who received a knee replacement were less than 69 years of age [33]. Unfortunately there are no precise official data regarding the TKA rates in the working age population in Poland. By comparison, the data published by The United Kingdom National Joint Registry show that, in 2016, patients up to 65 years of age accounted for 40% of cases [34]. Notably, the number of patients with diagnosed knee osteoarthritis is consistently growing. Similarly, the number of TKA procedures performed annually is increasing, and individuals up to 65 years of age are expected to be the largest group among those receiving this treatment [35].

In view of the above, and considering the fact that the duration of people’s working lives is increasing, RTW after TKA is an issue of great importance. However, in Poland there are no reliable data concerning this subject matter. For this reason, the present retrospective study was designed to determine the RTW rates after TKA in a Polish population, in order to provide reference data, and identify the factors impacting patients’ decisions to return to or resign from work, as well as their functional capacities following TKA. It was hypothesized that RTW rates in patients after TKA in Poland would be similar to those in developed countries.

Point 6: Materials methods / results : Clear

Response 6: Thank you very much for this comment.

Point 7: Discussion: better in the new version

Response 7: Thank you very much for this comment.

Point 8: conclusion Clear.

Response 8: Thank you very much for this comment.

Point 9: in Summery, new version is significantly better than old version, only abstract should be refined.

Response 9: Thank you very much for this comment. In accordance with the Reviewer’s recommendation, we have revised the abstract.

BEFORE

line 10-22: Abstract

1) Background: Total knee arthroplasty (TKA) performed on working-age patients significantly affects the participation of such patients in social life. A retrospective study was conducted to determine the return to work (RTW) rate after TKA. The goal of this study was to provide reference data for the Polish population and identify the factors impacting patients’ decisions to return to or resign from work, relative to their functional performance. 2) Methods: This retrospective study analyzed 431 patients. Ultimately, 48 individuals were enrolled. For this study, an interview on RTW was carried out. This interview also explored the factors impacting a patient’s decision to return to or resign from work. Functional performance was assessed using the Knee Outcome Survey Activities of Daily Living (KOS-ADL). 3) Results: Before TKA, 31.25% of those qualified for the study did not work and were receiving welfare benefits. After the surgery, 23 individuals did not return to work (47.9% of all those working prior to TKA). The group of those who did not work after TKA increased significantly, reaching 79.17%. 4) Conclusions: Considering the importance of returning to work for those who have not reached retirement age, further large-scale studies should be conducted while taking into account environmental factors and the quality of life among individuals after TKA.

AFTER

line 10-24: Abstract

1) Background: Total knee arthroplasty (TKA) performed on working-age patients significantly affects the participation of such patients in social life. A retrospective study was conducted to determine the return to work (RTW) rate after TKA. The goal of this study was to provide reference data for the Polish population and identify the factors impacting patients’ decisions to return to or resign from work, relative to their functional performance. 2) Methods: This retrospective involved 48 patients. An interview related to RTW was carried out to identify the factors impacting a patient’s decision to return to or resign from work. Functional performance was assessed using the Knee Outcome Survey Activities of Daily Living (KOS-ADL). 3) Results: Before TKA, 15 individuals (31.25%) qualified for the study did not work and were receiving welfare benefits. After the surgery, 23 individuals (47.9% of those working prior to TKA) did not return to work. The number of those who did not work after TKA increased to 38 (79.17%), which was a significant change. The mean level of functional performance after TKA assessed using KOS-ADL was 75.89. 4) Conclusions: The findings show that the rate of RTW after TKA in Poland is significantly lower than that in other countries. The reasons for this situation, as shown in the study, may be related to the lack of an occupational rehabilitation system, resulting in a paucity of information about the possibility to return to work and about opportunities for retraining.

Round 2

Reviewer 1 Report

Comments and Suggestions for Authors

Authors have made drastic changes to their work.

Author Response

Dear Reviewer,        

We thank you for re-reviewing our article titled, “Return to work after primary total knee arthroplasty: the first Polish retrospective study”. We thank the reviewer of all the comments.                                                                                                          

With kind regards,

The authors of the article

Point 1: Authors have made drastic changes to their work.

Response 1: Thank you for this valuable comment.

Reviewer 2 Report

Comments and Suggestions for Authors

This study has been revised to some extent, according to the recommendations. English language is readable. However, several points still remain not acceptable to revision changes, due to some aspects of the design of this study as authors have explained.

Title: brief and comprehensive, revised according to the recommendations.

Introduction

This session still remains extensive. Some paragraphs could be shortened, so interesting literature including kinematics to be mentioned.

Materials and methods

Study design has been more clearly presented, some points including scales used to assess healthy status have been clarified, while others not. Division of subgroups regarding related factors (including sex) is not feasible, due to study design.  

Discussion

Even if, this session has been revised still remains extensive and not focused on the explanation of the results taking into account the aims of this study.

Conclusions

Conclusions could emphasize on whether main aims of this study have been applied according to the design and what could be improved. However, this session mainly consists of general comments on lack of rehabilitation programs. Finally, even if, this study includes results, whisch are mainly not statistically important, which weakens following sessions and makes if of limited interest for the readers.

Author Response

Dear Reviewer,        

We thank you for re-reviewing our article titled, “Return to work after primary total knee arthroplasty: the first Polish retrospective study”. We have made every effort to improve our manuscript, as guided by the reviewer’s helpful suggestions.

We thank the reviewer of all the comments. Answers are summarised below. All changes are highlighted as red text in the manuscript.

We hope you will be pleased with the changes, and support the publication of our revised manuscript.

With kind regards,

The authors of the article

Point 1:

This study has been revised to some extent, according to the recommendations. English language is readable. However, several points still remain not acceptable to revision changes, due to some aspects of the design of this study as authors have explained.

Response 1: Thank you for this valuable comment. All the suggestions have been addressed in the revised manuscript.

Point 2: Title: brief and comprehensive, revised according to the recommendations.

Response 2: Thank you for the comment.

Point 3: Introduction

This session still remains extensive. Some paragraphs could be shortened, so interesting literature including kinematics to be mentioned.

Details regarding factors affecting on return to work have been added. However, these details make the introduction extended, taking into account that no literature analyzing knee replacement kinematics has been added (as suggested).

Response 3: Thank you for the helpful suggestions. The Introduction section has been carefully revised to address the comment; literature analyzing knee replacement kinematics has been added.

BEFORE

lines 28-99: Introduction

Knee osteoarthritis is a concern for increasingly more people. This affliction leads to significant impairment of functional capacities, which adversely affects activity and participation, gradually leading to a decrease in quality of life [1,2]. Total Knee Arthroplasty (TKA) is the most commonly applied intervention. This treatment is highly effective since it relieves pain and restores the function of the joint, consequently enabling improvements in the patients’ quality of life [3,4]. Although numerous studies have shown that TKA is one of the most successful and effective procedures in orthopedics [5-11], the outcomes can vary based on several factors, such as patient health status, including mental health, as well as expectations, comorbidities, age, preoperative pain ratings, preoperative levels of physical activity, and rehabilitation adherence [12-19]. Additionally, it was suggested that the design of post-TKA rehabilitation should take into account specific patient risk factors, including age, health status, social support status, and knee-related physical functioning before the surgery [20,21]. Furthermore, Styron showed that the factors linked to a faster return to work (RTW) after TKA included female gender, self-employment, better post-operative physical and mental health outcomes, a higher functional comorbidity index, and an accessible workplace accessible. A slower RTW after TKA was linked to the preoperative level of pain, more physically demanding work, and employee compensation [22]. Bardgett identified factors that affected the RTW process after TKA. According to patients, these factors included delays in surgical intervention, limited and often inconsistent RTW-related advice from healthcare professionals, and a lack of rehabilitation to optimize the patient's recovery and facilitate RTW [23].

According to the data from the National Health Fund, in Poland, as many as 37,821 knee replacement procedures were performed in 2020, including 30,615 TKA procedures. Primary knee arthroplasty was performed in 94.3% of the cases, and 95% of the patients were operated on due to primary gonarthrosis. Women accounted for 72% of all patients who received this treatment, and 80% of the patients were aged 60-79 years. On average, the female and male patients were aged 69 and 67 years, respectively. One in two patients who received a knee replacement in 2019 were 69 years of age or younger [24]. According to data published by The United Kingdom National Joint Registry, in 2016, patients up to 65 years of age accounted for 40% of cases [25]. Notably, the number of patients with diagnosed knee osteoarthritis is consistently growing. Similarly, the number of TKA procedures performed annually is increasing, and individuals up to 65 years of age are expected to be the largest group among those receiving this treatment [26].

Considering that the duration of people’s working lives is increasing, return to work (RTW) after TKA is an issue of great importance. Due to the scarcity of related research, the determinants of RTW after TKA are still unclear. It also remains difficult to compare and analyze the results because the definitions of work status vary widely, and different timeframes are often used to measure this status. In 2014, Tilbury presented the findings of a systematic review of studies investigating RTW after hip or knee replacement [27]. This review showed that the percentage of patients returning to work was between 25% and 95% in the period from 1 to 12 months after THA and between 71% and 83% from 3 to 6 months after TKA [27]. Furthermore, Van Leemput reported that the percentage of individuals working after TKA was between 36% and 89%. Those who worked before and returned to work after TKA accounted for 40% to 98% of the populations studied, whereas the mean duration of the rehabilitation period after TKA ranged from 7.7 to 16.6 weeks [28]. According to Van Leemput, a slower or no RTW after TKA was more likely to be observed in patients performing jobs of a more physical nature or among those who were absent from work before the surgery. However, most patients subjected to TKA returned to work after the procedure [28]. Leichtenberg established that a more advanced age, lower educational level, and preoperative absence from work were associated with a partial or no return to work [29]. More specifically, Leichtenberg reported that in a post-TKA group of 56 individuals aged below 65 years (mean 56 years), there were 40 individuals (71%) who worked prior to the surgery and returned to full-time work one year after the intervention, as well as 10 individuals (18%) who also returned to work but with less working time than before the intervention. Six patients (11%) did not return to work after TKA at one year following the surgery [29]. A study by Scott analyzed 289 patients after TKA. A multivariate analysis showed that heavy or moderate physical work and age were the main factors contributing to a return to employment of any kind or a return to the same job. In the group of patients working before TKA, all those below 50 years of age and 60% of those aged 50-54 years returned to work in some capacity [30]. According to the Nordic Arthroplasty Register, the number of TKA interventions doubled over a 15-year period, whereas the number of THA procedures increased by 2.5 times. Due to the strong results of this research, it was possible to expand the indications to younger patients of working age [31]. Patients also continued to work longer as the retirement age increased.

However, there are no reliable data on Polish patients. For this reason, the present retrospective study was designed to determine the RTW rates after TKA, provide reference data, and identify the factors impacting patients’ decisions to return to or resign from work, as well as their functional capacities following TKA. It was hypothesized that RTW rates in patients after TKA in Poland would be similar to those in developed countries.

AFTER

lines 28-88: Introduction

Knee osteoarthritis is a concern for increasingly more people. This affliction leads to significant impairment of functional capacities, which adversely affects activity and participation, gradually leading to a decrease in the quality of life [1,2]. Total Knee Arthroplasty (TKA) is the most commonly applied intervention. This treatment is highly effective since it relieves pain and restores the function of the joint, consequently enabling improvements in the patients’ quality of life [3,4]. The success of the intervention is often measured by how well a person can resume their normal activities,  and this may be linked to TKA kinematics and biomechanics. Poor kinematics and biomechanics after surgery can lead to issues such as pain, instability, and decreased range of motion, which can hinder a person's capability to perform various activities effectively. In contrast, proper alignment and functioning of the implant can improve the person's mobility and performance, increasing the likelihood of a successful recovery of various functions necessary for social participation [5-11]. Although numerous studies have shown that TKA is one of the most effective procedures in orthopedics [12-18], the outcomes can vary based on several factors, such as patient health status, including mental health, as well as expectations, comorbidities, age, preoperative pain ratings, preoperative levels of physical activity, and rehabilitation adherence [19-26].

Effectiveness of TKA can also be evaluated taking into account patients’ ability to return to work (RTW). In 2014, Tilbury in a systematic review the related literature showed that RTW rate at 3 to 6 months after TKA ranged between 71% and 83% [27]. Another systematic review, by Van Leemput, showed that RTW rate after TKA ranged from 36% to 89%. Those who worked before and returned to work after TKA accounted for 40% to 98% of the populations studied [28]. Given these significant differences in the reported statistics, it seems necessary to identify the factors impacting patients’ decisions about RTW after TKA. The study by Styron showed that a faster RTW after TKA was linked to female gender, self-employment, better post-operative physical and mental health outcomes, a higher functional comorbidity index, and an accessible workplace, whereas a slower RTW after TKA was related to the preoperative level of pain, more physically demanding work, and employee compensation [29]. Furthermore, according to Bardgett the RTW process after TKA was affected by delays in surgical intervention, limited and often inconsistent RTW-related advice from healthcare professionals, and a lack of rehabilitation to optimize the patient's recovery and facilitate RTW [30]. According to Van Leemput, a slower or no RTW after TKA was more likely to be observed in patients performing jobs of a more physical nature or among those who were absent from work before the surgery [28]. Leichtenberg established that a more advanced age, lower educational level, and preoperative absence from work were associated with a partial or no return to work [31]. A multivariate analysis of data collected from 289 patients by Scott et al. showed that heavy or moderate physical work and age were the main factors contributing to a return to employment of any kind or a return to the same job. In the group of patients working before TKA, all those younger than 50 years of age and 60% of those aged 50-54 years returned to work in some capacity [32].

According to the data from the National Health Fund, in Poland, as many as 37,821 knee replacement procedures were performed in 2020, including 30,615 TKA procedures. On average, the female and male patients were aged 69 and 67 years, respectively. Approximately 50% of those who received a knee replacement were less than 69 years of age [33]. Unfortunately there are no precise official data regarding the TKA rates in the working age population in Poland. By comparison, the data published by The United Kingdom National Joint Registry show that, in 2016, patients up to 65 years of age accounted for 40% of cases [34]. Notably, the number of patients with diagnosed knee osteoarthritis is consistently growing. Similarly, the number of TKA procedures performed annually is increasing, and individuals up to 65 years of age are expected to be the largest group among those receiving this treatment [35].

In view of the above, and considering the fact that the duration of people’s working lives is increasing, RTW after TKA is an issue of great importance. However, in Poland there are no reliable data concerning this subject matter. For this reason, the present retrospective study was designed to determine the RTW rates after TKA in a Polish population, in order to provide reference data, and identify the factors impacting patients’ decisions to return to or resign from work, as well as their functional capacities following TKA. It was hypothesized that RTW rates in patients after TKA in Poland would be similar to those in developed countries.

Point 4: it could be better to be shortened and probably incorporated in the first paragraph, to make introduction no so extended (hardly less than one page).

Response 4: Thank you for the helpful suggestions. We have revised the Introduction, and we have reorganised the paragraphs to make the text more clear.

BEFORE

lines 62-93: Introduction

Considering that the duration of people’s working lives is increasing, return to work (RTW) after TKA is an issue of great importance. Due to the scarcity of related research, the determinants of RTW after TKA are still unclear. It also remains difficult to compare and analyze the results because the definitions of work status vary widely, and different timeframes are often used to measure this status. In 2014, Tilbury presented the findings of a systematic review of studies investigating RTW after hip or knee replacement [27]. This review showed that the percentage of patients returning to work was between 25% and 95% in the period from 1 to 12 months after THA and between 71% and 83% from 3 to 6 months after TKA [27]. Furthermore, Van Leemput reported that the percentage of individuals working after TKA was between 36% and 89%. Those who worked before and returned to work after TKA accounted for 40% to 98% of the populations studied, whereas the mean duration of the rehabilitation period after TKA ranged from 7.7 to 16.6 weeks [28]. According to Van Leemput, a slower or no RTW after TKA was more likely to be observed in patients performing jobs of a more physical nature or among those who were absent from work before the surgery. However, most patients subjected to TKA returned to work after the procedure [28]. Leichtenberg established that a more advanced age, lower educational level, and preoperative absence from work were associated with a partial or no return to work [29]. More specifically, Leichtenberg reported that in a post-TKA group of 56 individuals aged below 65 years (mean 56 years), there were 40 individuals (71%) who worked prior to the surgery and returned to full-time work one year after the intervention, as well as 10 individuals (18%) who also returned to work but with less working time than before the intervention. Six patients (11%) did not return to work after TKA at one year following the surgery [29]. A study by Scott analyzed 289 patients after TKA. A multivariate analysis showed that heavy or moderate physical work and age were the main factors contributing to a return to employment of any kind or a return to the same job. In the group of patients working before TKA, all those below 50 years of age and 60% of those aged 50-54 years returned to work in some capacity [30]. According to the Nordic Arthroplasty Register, the number of TKA interventions doubled over a 15-year period, whereas the number of THA procedures increased by 2.5 times. Due to the strong results of this research, it was possible to expand the indications to younger patients of working age [31]. Patients also continued to work longer as the retirement age increased.

AFTER

lines 46-69: Introduction

Effectiveness of TKA can also be evaluated taking into account patients’ ability to return to work (RTW). In 2014, Tilbury in a systematic review the related literature showed that RTW rate at 3 to 6 months after TKA ranged between 71% and 83% [27]. Another systematic review, by Van Leemput, showed that RTW rate after TKA ranged from 36% to 89%. Those who worked before and returned to work after TKA accounted for 40% to 98% of the populations studied [28]. Given these significant differences in the reported statistics, it seems necessary to identify the factors impacting patients’ decisions about RTW after TKA. The study by Styron showed that a faster RTW after TKA was linked to female gender, self-employment, better post-operative physical and mental health outcomes, a higher functional comorbidity index, and an accessible workplace, whereas a slower RTW after TKA was related to the preoperative level of pain, more physically demanding work, and employee compensation [29]. Furthermore, according to Bardgett the RTW process after TKA was affected by delays in surgical intervention, limited and often inconsistent RTW-related advice from healthcare professionals, and a lack of rehabilitation to optimize the patient's recovery and facilitate RTW [30]. According to Van Leemput, a slower or no RTW after TKA was more likely to be observed in patients performing jobs of a more physical nature or among those who were absent from work before the surgery [28]. Leichtenberg established that a more advanced age, lower educational level, and preoperative absence from work were associated with a partial or no return to work [31]. A multivariate analysis of data collected from 289 patients by Scott et al. showed that heavy or moderate physical work and age were the main factors contributing to a return to employment of any kind or a return to the same job. In the group of patients working before TKA, all those younger than 50 years of age and 60% of those aged 50-54 years returned to work in some capacity [32].

Point 5: Materials and methods

Study design has been more clearly presented, some points including scales used to assess healthy status have been clarified, while others not. Division of subgroups regarding related factors (including sex) is not feasible, due to study design. 

Response 5: Thank you for this comment. Because of the research method and the main criterion of age the study group was small, and it was impossible to apply further division, e.g. based on sex.

Point 6: Materials and methods

No comments on BMI and height have been added. (It could be useful to include details even in this session and not in only in study limitations, making it clear to the readers soon after reading materials and methods session).

Response 6: Thank you for the helpful suggestions. This is an important issue. It is assumed that to be safely qualified for TKA, the patient should have BMI below 40, because of the lower risk of postoperative complications and shorter duration of the procedure. However, a number of studies show that BMI is not related to functional performance following TKA even 10 years after the procedure. Due to this BMI was not defined as an eligibility criterion in this study. Additionally, the participants’ current functional independence was verified on the day of the examination. The related information has been added in the Methods section.

[Li, H.; Gu, S.; Song, K.; Liu, Y.; Wang, J.; Wang, J.; Yin, Q. The influence of obesity on clinical outcomes following primary total knee arthroplasty: A prospective cohort study. Knee 2020, 27, 1057-1063.]

[Xu, S.;  Lim, W. A.J.; Chen, J.Y.  The influence of obesity on clinical outcomes of fixed-bearing unicompartmental knee arthroplasty: a ten-year follow-up study. Bone Joint J 2019, 101-B(2), 213-220.]

BEFORE

lines 113-117: Materials and Methods

Exclusion criteria included revision knee arthroplasty and simultaneous bilateral TKA, other musculoskeletal diseases (e.g., spinal stenosis), a history of other surgeries or internal diseases making it impossible to return to work, oncological diseases, musculoskeletal infections, and lack of consent to participate.

AFTER

lines 102-109: Materials and Methods

The exclusion criteria applied in the study included revision knee arthroplasty and simultaneous bilateral TKA, other musculoskeletal diseases (e.g., spinal stenosis), a history of other surgeries or internal diseases making it impossible to return to work, oncological diseases, musculoskeletal infections, and lack of consent to participate. A lack of functional independence on the day of the examination was an excluding factor. The participant’s Body Mass Index (BMI) was calculated based on his/her body weight and height on the day of the examination. The BMI value did not impact the participants’ eligibility to be enrolled for the study [36,37].

Point 7: Materials and methods

Surgical procedures and type of endoprostheses are better defined, according to the recommendations.

Response 7: Thank you for the comment.

Point 8: Results

Flow diagram should be easily read.(sentences could be even more brief).

Response 8: Thank you for the helpful suggestions. We have  revised the Flow diagram accordingly.

Point 9: Discussion

Even if, this session has been revised still remains extensive and not focused on the explanation of the results taking into account the aims of this study.

Response 9: Thank you for the helpful suggestions. We have revised the discussion, and we have removed parts of the text, as suggested. We have revised first sentence in the Study Limitations.

BEFORE

lines 271-444: Discussion

The present study asked three questions: How many people return to work after TKA one year after surgery, what factors contribute to the decision to resign from work, and what is the level of functional performance among the study participants?

The initial assessment involved a large group of 431 patients who received unilateral TKA due to knee osteoarthritis. It was shown that as many as 85.6% of the individuals in the group did not meet the age criterion. The vast majority of patients had reached the retirement age by the day of the surgery. The participants’ ages on the day of TKA agreed with the data in the National Joint Registry (NJR), where those aged 50-59 years in 2020 accounted for 15% of 53,782 total individuals after TKA [25]. This is an important issue due to increasing life expectancy and the need for individuals to continue working in order to maintain quality of life in both material and social terms. However, the present study did not answer an important question, namely, how many of those excluded from the study had discontinued their gainful employment and started receiving welfare benefits due to osteoarthritis before reaching retirement age. This is important in the context of the selected timing of the intervention and the potential return to work.

There are few studies focusing on RTW after TKA. The average time to return to work was found to be between 8 and 12 weeks following TKA [37,22], whereas the RTW rates were between 40% and 98% in the case of TKA [38]. Mongin investigated the RTW rates at one year following TKA or THA in a group of 299 individuals below 65 years of age. In that group, 144 individuals worked until the day of the surgery. After TKA, an average of 38 patients returned to work (52.7%) after 141±100.69 days, and 34 did not resume work [39]. In the present study, 31.25% of the individuals included did not work before the surgery. Those not working were receiving welfare benefits (a pension) due to a degenerative joint disease of the knee. The group of those who did not work after TKA increased significantly to a level of 79.17%. All non-working participants were receiving a pension because they were unable to perform their previous job and presented a low level of fitness. Only 20.83% of participants, therefore, returned to work. This is a very low percentage compared to the results of many studies in which RTW rates after TKA were between 40% and 98%.

The present study provides quantitative information on the RTW rates after TKA. To date, very few studies in Poland have reported data related to RTW after hip replacement [40], and no studies have focused on RTW after TKA. A retrospective study by Foote involving 109 individuals under 60 years of age showed that 82% of patients who had worked before surgery and received patellofemoral replacement or unicompartmental replacement returned to work within 12 weeks. Following TKA, 54% of the patients returned to work within 20 weeks after surgery [35]. In the present study, the RTW rate after TKA was 30.3%. Before TKA, 31.25% of the patients were receiving a pension due to osteoarthritis of the knee. After the intervention, 79.16% were receiving a pension due to the disease and their post-operative condition. The low RTW rate in the study group may be due to a number of reasons. Individuals performing physical jobs, including agricultural workers, accounted for 62.5% of the group. Those who did not return to work after TKA noted that their main reason for resigning from work was the inability to perform the job they had before the surgery and the lack of opportunities to retrain or find another job (58.8%). Poor self-assessed health after TKA was the second most important reason for giving up work and was reported by 23.5% of the participants. Only two participants (11.8%) reported that the opportunity to receive welfare benefits was their reason for giving up work, whereas 5.9% of the participants reported a lack of acceptance from their employer as the reason. When asked whether they were informed about the possibility of retraining, the participants who provided answers concerningly reported that they had not been given any such information before or after TKA. Nevertheless, even if certain conditions were met, such as opportunities for retraining or greater employer acceptance, the majority of the participants stated that they would not take a job after TKA. In addition, poor health after TKA was reported by 23.5% of the participants as a reason to resign from work. Achieving the highest possible level of functional capacity after TKA is the main goal of therapy. We hypothesized that individuals who returned to work after TKA would present a higher level of functional performance than those who did not return to work after TKA. However, our findings presented a different picture. Those who did not work reported fewer limitations due to pain, swelling, and joint stiffness, as well as fewer limitations related to gait and basic gait function, compared to those who returned to work. Overall, the difference in functional performance between the two groups was not statistically significant. Therefore, other factors may have impacted the decision to return to or resign from work among the study group. Since no assessment with KOS-ADL was performed before TKA, it was impossible to assess the effects of changes in functional performance on RTW after surgery. However, there are conflicting opinions about the relationship between functional performance and RTW after TKA. Leichtenberg assessed pre-operative functional performance using KOS-ADL and reported a positive relationship between the KOS-ADL score and RTW after TKA [29]. Conversely, Kuijer did not find a significant relationship between the KOS-ADL score and RTW [41]. In the group of individuals who returned to work, those performing office jobs presented a lower rating of functional performance than that reported by patients who performed physical work in agriculture or other sectors. Consequently, there are no grounds for claiming that physical activity, including rehabilitation, impacted the perception of functional performance in the study group. However, there remains a need for further research to reliably assess the impact of rehabilitation on RTW after TKA.

Currently, there is no consistent process for optimizing RTW among working-age patients after TKA. The effects of late surgical intervention and absent or limited RTW-related counselling contribute to potential delays in return to work [42-44]. Therefore, it is necessary to refocus healthcare provisions for this group of patients, particularly with regard to RTW-oriented rehabilitation. Such provisions could inform patients about the possibility to perform work and contain tailored interventions to optimize patient outcomes [23]. A previous study showed that the process of recovery after TKA might benefit from high-volume exercise applied before surgery. However, assessments of exercise-based interventions applied after surgery did not show the advantage of one exercise program over another [45]. Therefore, there is no clear evidence that rehabilitation after TKA based on a specific type of activity would generally enhance recovery in patients [46,47]. It appears that earlier and more intense physical rehabilitation programs involving exercise (e.g., strength training) produce better effects than less intense programs (e.g., ADL training without strength training) [48]. The most recent reports concerning the impacts of various rehabilitation strategies on RTW outcomes suggest that specific rehabilitation interventions should primarily consider patients' expectations for their post-operative recovery, their willingness to engage in exercise and physical activity, and their pre-operative functional performance [45]. Therefore, future strategies should emphasize the importance of pre-operative exercises (pre-habilitation) in patients with reduced pre-operative knee extension strength, impaired functional performance, and a tendency to fall [45].

Patients taking part in this study indicated a need for counselling related to finding work that would match their specific employment requirements. Participants noted that such information could potentially prompt a return to work but also admitted that they did not want to return to work after TKA. The study group mainly included physical workers and agricultural workers, i.e., individuals with limited options for retraining who were also unable to continue to perform physically demanding jobs due to the surgery.

It is also necessary to emphasize the importance of the welfare benefits that most of the patients received after TKA since this factor seemed to encourage patients to withdraw from the labor market.

In reference to studies carried out to date in various countries, Tilbury noted that the timing of RTW may be affected by the health care and welfare security systems. As an example, Tilbury reported that sick leave from work in the Netherlands is paid in full for the first two years [49]. Lombardi noted that in the USA and other countries, affordable health insurance is an advantage, highlighting a study in which 77% of individuals aged 18-69 years post-TKA obtained benefits from the health insurance provided by their employer [38]. The low RTW rate identified in this study may be linked to the Polish healthcare and social security system. In Poland, welfare benefits are commonly awarded to individuals incapable of working due to a degenerative disease or surgery. Given the relationship between minimum wage and the amount of these benefits, welfare may be an acceptable solution for many people, especially physical workers with low occupational qualifications. Lombardi suggested that one of the main reasons for not returning to work after TKA is retirement and the use of disability benefits prior to surgery [38]. The present findings support this opinion. As many as 31% of the working-age participants in this study received benefits even before the surgery. After TKA, this group increased significantly.

The presented findings highlighted factors affecting RTW after TKA that are consistent with those reported in other studies. To answer the main question, the group of participants was narrowed to those who were of working age according to Polish regulations and had received surgery within the last year. Unfortunately, the present findings differ significantly from those most frequently reported in other studies. This difference may be due to the small number of participants, but it should be noted that as many as 85% of sampled individuals did not meet the age criterion. The reasons for the low rate of RTW in the study group included a lack of information about the possibility to return to work after surgery, various limitations, and types of work. In Poland, the system of occupational rehabilitation is insufficiently developed. Information on opportunities for retraining or finding another job is not made available to patients with developing osteoarthritis or during the pre-operative period. The most common route for the treatment and rehabilitation process is surgery; rehabilitation; satisfactory recovery of functional capacities; and, as demonstrated here, the acquisition of an incapacity pension. This process was confirmed by the findings of our study, which showed that functional performance after TKA was similar between those who returned to work and those that did not.

Considering the growing number of knee replacement surgeries performed on working-age patients, as well as the increasing life expectancy and corresponding need to increase the duration of one’s working life, this group of patients should receive coordinated medical and rehabilitation care. Considering the importance of returning to work among those who have not yet reached retirement age, further large-scale studies should be conducted considering environmental factors and the quality of life among individuals after TKA. It is also necessary to conduct research evaluating the effectiveness of interventions applied in occupational rehabilitation and functional training after TKA. Over the past few years, high hopes have been placed on the modification of surgical technology using computer navigation and robotics. Indeed, new techniques may reduce the number of patients reporting post-operative problems that prevent them from returning to work. Successful RTW is increasingly recognized as an important outcome for this group, with social and economic consequences for patients, employers, and society.

Study limitations

Despite the retrospective recruitment of 431 patients, only 48 individuals met the inclusion criteria, making further analysis difficult. A multicenter study taking into account different social settings is essential to obtain an adequate overview of the problem. The study also did not assess the working hours of those who chose to return to work, so the findings did not indicate whether these participants returned to work full-time or part-time. Another limitation is the fact that no information was acquired on the participants’ rehabilitation process. Indeed, rehabilitation can vary greatly, ranging from the most basic post-operative therapies to in-patient programs at hospital rehabilitation units or systematic treatment programs at rehabilitation centers. Thus, the type of rehabilitation received may be related to one’s decision to return to work. Furthermore, the eligibility criteria in the present study did not include the BMI or height of the patients. One’s BMI and height can considerably impact the recovery of functional capacities after TKA. Thus, these factors should be considered in further research. Future studies should also distinguish between additional sex-related subgroups due to the different age thresholds applied. The division of groups by age and sex is very important, but our sample was narrowed to individuals with primary and unilateral TKA due to degenerative disease, leaving only 48 individuals for the final analysis. For this reason, a further division into subgroups would have made the process of drawing conclusions more difficult.

AFTER

lines 264-422: Discussion

The present study asked three questions, i.e., How many people return to work after TKA one year after surgery, what factors contribute to the decision to resign from work, and what is the level of functional performance among the study participants?

The initial group of potential study participants comprised 431 patients who received unilateral TKA due to knee osteoarthritis. As many as 85.6% of these patients had reached the retirement age by the day of the surgery, which means that nearly 15% of those after TKA were of working age, in line with data in the UK National Joint Registry where those aged 50-59 years in 2020 accounted for 15% of 53,782 total individuals after TKA [34]. This is an important issue due to increasing life expectancy and the need for individuals to continue working in order to maintain quality of life in both material and social terms. However, the present study did not answer an important question, namely, how many of those excluded from the study had discontinued their gainful employment and started receiving welfare benefits due to osteoarthritis before reaching retirement age. This is important in the context of the selected timing of the intervention and the potential return to work.

In the present study, 31.25% of the participants did not work before the surgery, and they were receiving welfare benefits (a pension) due to a degenerative joint disease of the knee. The group of those who did not work after TKA increased significantly to a level of 79.17%. All non-working participants were receiving a pension because they were unable to perform their previous job and presented a low level of fitness. Only 20.83% of participants, therefore, returned to work, which is a very low rate compared to 40%-98% reported in other studies [43], with the average time to return to work between 8 and 12 weeks after TKA [44,29]. Mongin investigated the RTW rates at one year following TKA or Total Hip Arthroplasty in a group of 241 individuals below 65 years of age. In that group, 144 individuals worked until the day of the surgery. After TKA or THA, an average of 38 patients returned to work (52.7%) after 141±100.69 days, and 34 did not resume work [45].

The present study provides quantitative information on the RTW rates after TKA. To date, very few studies in Poland have reported data related to RTW after hip replacement [46], and no studies have focused on RTW after TKA. A retrospective study by Foote involving 109 individuals under 60 years of age showed that 82% of patients who had worked before surgery and received patellofemoral replacement or unicompartmental replacement returned to work within 12 weeks. Following TKA, 54% of the patients returned to work within 20 weeks after surgery [41]. In the present study, the RTW rate after TKA was 30.3%. Before TKA, 31.25% of the patients were receiving a pension due to osteoarthritis of the knee, whereas after the intervention 79.16% were receiving a pension due to the disease and their post-operative condition. The low RTW rate in the study group may be due to a number of reasons. Individuals performing physical jobs, including agricultural workers, accounted for 62.5% of the group. Those who did not return to work after TKA noted that their main reason for resigning from work was the inability to perform the job they had before the surgery and the lack of opportunities to retrain or find another job (58.8%). Poor self-assessed health after TKA was the second most important reason for giving up work and was reported by 23.5% of the participants. Only two participants (11.8%) reported that the opportunity to receive welfare benefits was their reason for giving up work, whereas 5.9% of the participants reported a lack of acceptance from their employer as the reason. When asked whether they were informed about the possibility of retraining, the participants who provided answers concerningly reported that they had not been given any such information before or after TKA. Nevertheless, even if certain conditions were met, such as opportunities for retraining or greater employer acceptance, the majority of the participants stated that they would not take a job after TKA. Achieving the highest possible level of functional capacity after TKA is the main goal of therapy. We hypothesized that individuals who returned to work after TKA would present a higher level of functional performance than those who did not return to work after TKA. However, our findings presented a different picture. Those who did not work reported fewer limitations due to pain, swelling, or joint stiffness, and fewer limitations related to gait and basic gait function, compared to those who returned to work. Since the difference in functional performance between the two groups was not statistically significant, other factors may have impacted RTW rate in the study group. No pre-operative scores in KOS-ADL are available, therefore the effects of changes in post-operative functional performance on RTW could not be assessed. However, there are conflicting opinions about the relationship between functional performance and RTW after TKA. Leichtenberg reported a positive relationship between the pre-operative KOS-ADL score and RTW after TKA [31], whereas Kuijer found no such relationship between these two factors [47]. In the group of individuals who returned to work, those performing office jobs presented a lower rating of functional performance than that reported by patients who performed physical work in agriculture or other sectors. Consequently, there are no grounds for claiming that physical activity, including rehabilitation, impacted the perception of functional performance in the study group. However, there remains a need for further research to reliably assess the impact of rehabilitation on RTW after TKA.

Currently, there is no consistent process for optimizing RTW among working-age patients after TKA. The effects of late surgical intervention and absent or limited RTW-related counselling contribute to potential delays in return to work [48-50]. Therefore, it is necessary to refocus healthcare provisions for this group of patients, particularly with regard to RTW-oriented rehabilitation. Such provisions could inform patients about the possibility to perform work and contain tailored interventions to optimize patient outcomes [30]. A previous study showed that the process of recovery after TKA might benefit from high-volume exercise applied before surgery. However, assessments of exercise-based interventions applied after surgery did not show the advantage of one exercise program over another [51]. Therefore, there is no clear evidence that rehabilitation after TKA based on a specific type of activity would generally enhance recovery in patients [52,53]. It appears that earlier and more intense physical rehabilitation programs involving exercise (e.g., strength training) produce better effects than less intense programs (e.g., ADL training without strength training) [54]. The most recent reports concerning the impacts of various rehabilitation strategies on RTW outcomes suggest that specific rehabilitation interventions should primarily consider patients' expectations for their post-operative recovery, their willingness to engage in exercise and physical activity, and their pre-operative functional performance [51]. Therefore, future strategies should emphasize the importance of pre-operative exercises (pre-habilitation) in patients with reduced pre-operative knee extension strength, impaired functional performance, and a tendency to fall [51].

In reference to studies carried out to date in various countries, Tilbury noted that the timing of RTW may be affected by the health care and welfare security systems. As an example, Tilbury reported that sick leave from work in the Netherlands is paid in full for the first two years [55]. Lombardi noted that in the USA and other countries, affordable health insurance is an advantage, highlighting a study in which 77% of individuals aged 18-69 years post-TKA obtained benefits from the health insurance provided by their employer [38]. The low RTW rate identified in this study may be linked to the Polish healthcare and social security system. In Poland, welfare benefits are commonly awarded to individuals incapable of working due to a degenerative disease or surgery. Given the relationship between minimum wage and the amount of these benefits, welfare may be an acceptable solution for many people, especially physical workers with low occupational qualifications. Lombardi suggested that one of the main reasons for not returning to work after TKA is retirement and the use of disability benefits prior to surgery [43]. The present findings support this opinion. As many as 31% of the working-age participants in this study received benefits even before the surgery. After TKA, this group increased significantly.

The presented findings highlighted factors affecting RTW after TKA that are consistent with those reported in other studies. To answer the main question, the group of participants was narrowed to those who were of working age according to Polish regulations and had received surgery within the last year. Unfortunately, the present findings differ significantly from those most frequently reported in other studies. This difference may be due to the small number of participants, but it should be noted that as many as 85% of sampled individuals did not meet the age criterion. The reasons for the low rate of RTW in the study group included a lack of information about the possibility to return to work after surgery, various limitations, and types of work. In Poland, the system of occupational rehabilitation is insufficiently developed. Information on opportunities for retraining or finding another job is not made available to patients with developing osteoarthritis or during the pre-operative period. The most common route for the treatment and rehabilitation process is surgery; rehabilitation; satisfactory recovery of functional capacities; and, as demonstrated here, the acquisition of an incapacity pension. This process was confirmed by the findings of our study, which showed that functional performance after TKA was similar between those who returned to work and those that did not.

Considering the growing number of knee replacement surgeries performed on working-age patients, as well as the increasing life expectancy and corresponding need to increase the duration of one’s working life, this group of patients should receive coordinated medical and rehabilitation care. Considering the importance of returning to work among those who have not yet reached retirement age, further large-scale studies should be conducted considering environmental factors and the quality of life among individuals after TKA. It is also necessary to conduct research evaluating the effectiveness of interventions applied in occupational rehabilitation and functional training after TKA. Over the past few years, high hopes have been placed on the modification of surgical technology using computer navigation and robotics. Indeed, new techniques may reduce the number of patients reporting post-operative problems that prevent them from returning to work. Successful RTW is increasingly recognized as an important outcome for this group, with social and economic consequences for patients, employers, and society.

Study limitations

The group of participants was small as a result of the adopted age criterion. It would be necessary to change the age limit to 60 and 65 years in women and men respectively or to conduct a multicenter study taking into account different social settings in order to obtain an adequate overview of the problem. The study also did not assess the working hours of those who chose to return to work, so the findings did not indicate whether these participants returned to work full-time or part-time. Another limitation is the fact that no information was acquired on the participants’ rehabilitation process. Indeed, rehabilitation can vary greatly, ranging from the most basic post-operative therapies to in-patient programs at hospital rehabilitation units or systematic treatment programs at rehabilitation centers. Thus, the type of rehabilitation received may be related to one’s decision to return to work. Furthermore, the eligibility criteria in the present study did not include the BMI or height of the patients. One’s BMI and height can considerably impact the recovery of functional capacities after TKA. Thus, these factors should be considered in further research. Future studies should also distinguish between additional sex-related subgroups due to the different age thresholds applied. The division of groups by age and sex is very important, but our sample was narrowed to individuals with primary and unilateral TKA due to degenerative disease, leaving only 48 individuals for the final analysis. For this reason, a further division into subgroups would have made the process of drawing conclusions more difficult.

Point 10: Discussion

Lack of KOS-ADL has been explained extensively explained (could be shortened)

Response 10: Thank you for the helpful suggestions. The discussion has been revised and shortened, as suggested.

BEFORE

lines 325-346: Discussion

Achieving the highest possible level of functional capacity after TKA is the main goal of therapy. We hypothesized that individuals who returned to work after TKA would present a higher level of functional performance than those who did not return to work after TKA. However, our findings presented a different picture. Those who did not work reported fewer limitations due to pain, swelling, and joint stiffness, as well as fewer limitations related to gait and basic gait function, compared to those who returned to work. Overall, the difference in functional performance between the two groups was not statistically significant. Therefore, other factors may have impacted the decision to return to or resign from work among the study group. Since no assessment with KOS-ADL was performed before TKA, it was impossible to assess the effects of changes in functional performance on RTW after surgery. However, there are conflicting opinions about the relationship between functional performance and RTW after TKA. Leichtenberg assessed pre-operative functional performance using KOS-ADL and reported a positive relationship between the KOS-ADL score and RTW after TKA [29]. Conversely, Kuijer did not find a significant relationship between the KOS-ADL score and RTW [41]. In the group of individuals who returned to work, those performing office jobs presented a lower rating of functional performance than that reported by patients who performed physical work in agriculture or other sectors. Consequently, there are no grounds for claiming that physical activity, including rehabilitation, impacted the perception of functional performance in the study group. However, there remains a need for further research to reliably assess the impact of rehabilitation on RTW after TKA.

AFTER

lines 315-334: Discussion

Achieving the highest possible level of functional capacity after TKA is the main goal of therapy. We hypothesized that individuals who returned to work after TKA would present a higher level of functional performance than those who did not return to work after TKA. However, our findings presented a different picture. Those who did not work reported fewer limitations due to pain, swelling, or joint stiffness, and fewer limitations related to gait and basic gait function, compared to those who returned to work. Since the difference in functional performance between the two groups was not statistically significant, other factors may have impacted RTW rate in the study group. No pre-operative scores in KOS-ADL are available, therefore the effects of changes in post-operative functional performance on RTW could not be assessed. However, there are conflicting opinions about the relationship between functional performance and RTW after TKA. Leichtenberg reported a positive relationship between the pre-operative KOS-ADL score and RTW after TKA [31], whereas Kuijer found no such relationship between these two factors [47]. In the group of individuals who returned to work, those performing office jobs presented a lower rating of functional performance than that reported by patients who performed physical work in agriculture or other sectors. Consequently, there are no grounds for claiming that physical activity, including rehabilitation, impacted the perception of functional performance in the study group. However, there remains a need for further research to reliably assess the impact of rehabilitation on RTW after TKA.

Point 8: Conclusions

Conclusions could emphasize on whether main aims of this study have been applied according to the design and what could be improved. However, this session mainly consists of general comments on lack of rehabilitation programs. Finally, even if, this study includes results, whisch are mainly not statistically important, which weakens following sessions and makes if of limited interest for the readers.

Response 8: Thank you for the comment. We have revised the Conclusion section in line with the Reviewer’s suggestion, to emphasize the findings in relation to the main aims of this study.

BEFORE

lines 446-455: Conclusions

This study was conducted due to the increasing number of patients receiving TKA for knee osteoarthritis and the associated medical and social problems. The findings show that the rate of RTW after TKA in Poland is significantly lower than that in other countries. The reasons for this situation, as shown in the study, may be related to the lack of an occupational rehabilitation system, resulting in a paucity of information about the possibility to return to work and opportunities for retraining. The results of this study suggest a significant deficit in occupational rehabilitation programs designed for individuals who have regained a significant degree of functional capacity after TKA due to knee osteoarthritis.

AFTER

lines 424-436: Conclusions

The findings suggest that the rate of RTW after TKA in Poland is significantly lower than that in other countries. The reasons for this situation, as shown in the study, may be related to the lack of an occupational rehabilitation system, resulting in a paucity of information about the possibility to return to work and opportunities for retraining. Patients taking part in this study indicated a need for counselling related to finding work that would match their specific employment requirements. Participants noted that such information could potentially prompt a return to work but also admitted that they did not want to return to work after TKA. The study group mainly included physical workers and agricultural workers, i.e., individuals with limited options for retraining who were also unable to continue to perform physically demanding jobs due to the surgery. It is also necessary to emphasize the importance of the welfare benefits that most of the patients received after TKA since this factor seemed to encourage patients to withdraw from the labor market.
